# Temporal Straightening for Latent Planning

Ying Wang[1]   Oumayma Bounou[1]   Gaoyue Zhou[1]   Randall Balestriero[2]   Tim G. J. Rudner[3]
Yann LeCun[*1]   Mengye Ren[*1]

## Abstract

Learning good representations is essential for latent planning with world models. While pretrained visual encoders produce strong semantic visual features, they are not tailored to planning and contain information irrelevant—or even detrimental—to planning. Inspired by the perceptual straightening hypothesis in human visual processing, we introduce temporal straightening to improve representation learning for latent planning. Using a curvature regularizer that encourages locally straightened latent trajectories, we jointly learn an encoder and a predictor of a Joint-Embedding Predictive Architecture (JEPA) world model. We show that reducing curvature this way makes the Euclidean distance in latent space a better proxy for the geodesic distance and improves the conditioning of the planning objective. We demonstrate empirically that temporal straightening makes gradient-based planning more stable and yields significantly higher success rates across a suite of goal-reaching tasks. Our code is in https://agenticlearning.ai/temporal-straightening/.

## 1. Introduction

Latent world models offer a compelling solution for planning due to better efficiency and generalization (Nguyen & Widrow, 1989; Sutton, 1991; Ha & Schmidhuber, 2018; Hafner et al., 2020; 2021; 2023; Hansen et al., 2022; 2024). They compress high-dimensional observations into compact latent representations, learn predictive dynamics in that latent space, and enable imaginary rollouts for action optimization. Compared to operating directly in pixel or state space, the latent abstraction reduces dimensionality and ig-

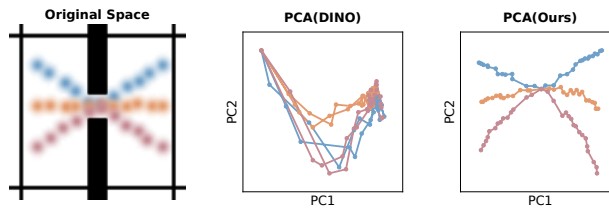

*Figure 1.* Latent trajectories encoded by a pretrained visual encoder are usually highly curved, increasing the difficulty of prediction and planning. We learn a representation space where feasible trajectories are straighter to facilitate latent planning.

nores noise, making dynamics learning more efficient. At test time, planning is typically posed as optimizing an action sequence by rolling the model forward and minimizing a cost function between the goal and the predicted states in the latent space.

In practice, however, optimization in the learned latent space remains challenging. The induced planning objective is typically highly non-convex, potentially causing gradient-based optimizers to struggle. As a result, many successful practices (Hafner et al., 2019; Hansen et al., 2024; Zhou et al., 2025; Sobal et al., 2025; Terver et al., 2025) rely on search-based methods such as CEM (Rubinstein, 1997) or MPPI (Williams et al., 2015), which achieve competitive performance but introduce a substantial compute burden and latency. Moreover, commonly used goal cost metrics based on Euclidean distance can be misleading if the embedding space is not properly regularized. In particular, when latent trajectories are highly curved, straight-line distances in embedding space misrepresent the geodesic distance along feasible transitions. These challenges call for better representations that facilitate latent planning.

What is a "good" representation for latent planning? Although general-purpose visual pretraining provides powerful semantic-aware features, it is not tailored to the dynamics of the environment and often retains plenty of planning-irrelevant low-level details. We argue that planning could benefit from representations that are (i) sufficient for predicting dynamics but without task-irrelevant information and (ii) properly regularized so that embedding distances reflect the geodesic distance and gradient-based optimization is reliable. With such representations, we can exploit the differentiability of latent world models and enable efficient

*Equal advising [1]New York University [2]Brown University [3]University of Toronto. Correspondence to: Ying Wang <yw3076@nyu.edu>, Yann LeCun <yann.lecun@nyu.edu>, Mengye Ren <mengye@nyu.edu>.

*Proceedings of the 43rd International Conference on Machine Learning*, Seoul, South Korea. PMLR 306, 2026. Copyright 2026 by the author(s).

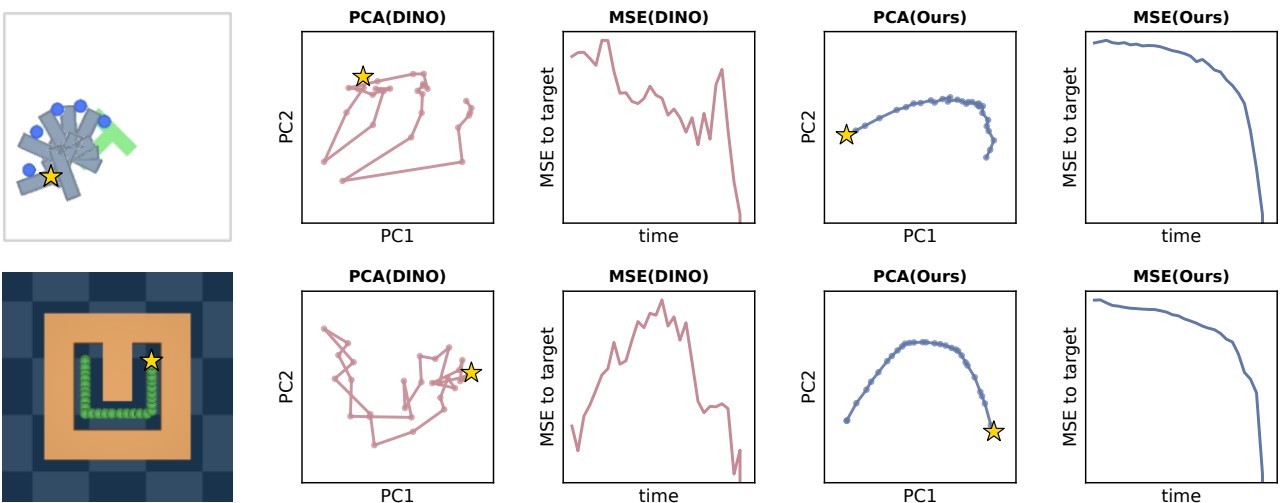

*Figure 2.* Latent trajectories before vs. after straightening. The upper PushT example is a rotation and the bottom UMaze example shows the agent traveling from the left top to the right top, with the star denoting the target. Straightening yields less curved and smoother trajectories, and makes Euclidean distance a more faithful proxy for geodesic progress towards the goal. More examples are in Section E.2.

gradient-based planning, bypassing the need for computationally expensive search-based methods.

Inspired by the perceptual straightening hypothesis in human vision (Hénaff et al., 2019), which posits that visual systems transform complex natural videos into straighter internal representations, we introduce a simple approach to straighten latent trajectories for planning. Concretely, we jointly learn an encoder and a predictor of a Joint-Embedding Predictive Architecture (JEPA) world model, while imposing regularization on the curvature of latent trajectories during training. We find that the JEPA prediction objective alone induces *implicit straightening* to some extent, and introducing an *explicit* curvature regularizer further strengthens and stabilizes this effect. The resulting encoded trajectories are significantly straighter, with Euclidean distances better aligned with geodesic distances (Figure 2). We prove that reducing curvature improves convergence of gradient-based planners, and observe superior empirical gains across a suite of goal-reaching tasks: open-loop planning success improves by 20–60% and MPC by 20-30% with a simple gradient-based planner.

## 2. Related Work

While early visual world models directly predict in pixel spaces and use generated images for control (Oh et al., 2015; Finn & Levine, 2017; Ebert et al., 2018; Du et al., 2023), an increasing number of recent works first encode high-dimensional sensory inputs into compact latent representations and plan in the resulting latent space. Learning representations is central to these latent world models.

To obtain meaningful representations for world modeling,

prior methods add reconstruction-based objectives when training the encoder along with the predictor (Watter et al., 2015; Zhang et al., 2019b; Levine et al., 2020; Ha & Schmidhuber, 2018; Hafner et al., 2019; 2020; Micheli et al., 2023; Robine et al., 2023). However, these reconstruction objectives overemphasize low-level visual details that are unnecessary for planning and may fail to capture task-relevant information. More recent approaches decouple perception from dynamics by leveraging strong pretrained visual encoders (Nair et al., 2022; Zhou et al., 2025; Bar et al., 2025; Goswami et al., 2025; Bai et al., 2025; Assran et al., 2025). Closest to our setup, DINO-WM (Zhou et al., 2025) trains task-agnostic predictors and plans directly in frozen DINOv2 (Oquab et al., 2024) feature space. While DINOv2 features provide high-quality visual representations, they are not optimized for planning and may lead to planning objectives that are challenging to optimize. In this work, we improve the representation space for planning by introducing a curvature regularization during world model training.

Joint-Embedding Predictive Architecture (JEPA) emerges as a promising paradigm for world models by learning representations through prediction (LeCun, 2022; Bardes et al., 2024; Assran et al., 2025). It aims to capture predictable structure without retaining unpredictable low-level details, making it more effective and efficient than reconstruction objectives (Assel et al., 2025). This paradigm has been shown to be effective for predictive modeling and planning, with training from scratch on offline simulator data (Sobal et al., 2025) and large-scale real-world video pretraining followed by action-conditioned post-training for robotic planning (Assran et al., 2025). Our work also belongs to the JEPA family and focuses on learning better representations.

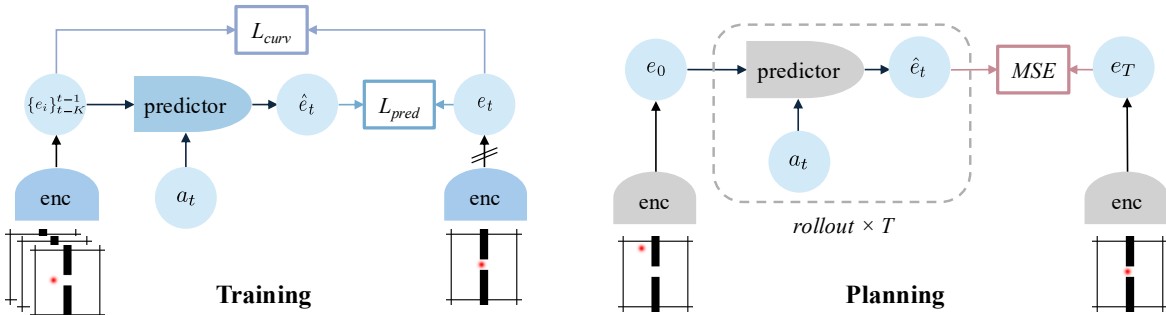

*Figure 3.* During training, we minimize the prediction loss between the predicted embedding $\hat{z}_t^t$ and the target $z_t^t$ with stop-grad in the target branch, and minimize the local curvature of embeddings. During planning, we rollout for the horizon $T$ using the trained predictor and select optimal actions that minimize the cost between the predicted terminal state and the goal in the embedding space.

Temporal Contrastive Learning (Sermanet et al., 2018; Dave et al., 2022; Eysenbach et al., 2024; Yang & Ren, 2025) is also a popular paradigm for learning representations that can reflect the temporal relationships. It encourages temporally close frames to have similar embeddings while distant frames have more dissimilar embeddings through InfoNCE loss (Radford et al., 2021). However, how to choose positive and negative samples requires careful tuning and this objective might push away geodesically close states if suboptimal trajectories are used. Instead, our regularization-based method doesn't require negatives and applies *local* straightening without requiring expert trajectories.

Motivated by the perceptual straightening hypothesis in human vision (Hénaff et al., 2019), some prior works have examined implicit straightening in pretrained visual encoders (Harrington et al., 2023; Internò et al., 2025) or used it as an objective to obtain robust video models (Niu et al., 2024; Bagad & Zisserman, 2025). Relatedly, early work on learning linearized video representations also regularizes the curvature of latent trajectories (Goroshin et al., 2015). Implicit straightening is also observed in autoregressive language models optimized for next-word prediction (Hosseini & Fedorenko, 2023; Hosseini et al., 2026). To the best of our knowledge, however, none of these prior works have studied the impact of straightening on world modeling and planning. We further discuss the connection of our work to a broader literature in control and learning plannable representations in Section D.

## 3. Temporal Straightening

We consider control tasks with high-dimensional observations $o_t \in \mathbb{R}^{n_o}$ of an agent interacting with its environment using actions $a_t \in \mathbb{R}^{n_a}$. Our goal is to learn a world model that maps observations to a latent space and models the dynamics in this space, which we use for latent planning.

In this section, we first outline the architecture of our world model, then define the training objectives with a novel geometric regularization that straightens latent trajectories.

### 3.1. World model

Our world model predicts future states in a learned latent space and consists of three components: a sensory encoder, an action encoder, and a predictor.

**Sensory encoder.** The sensory encoder $\mathcal{E}_\phi^s$ maps raw observations $o_t$ into latent representations

$$z_t \in \mathbb{R}^d = \mathcal{E}_\phi^s(o_t). \tag{1}$$

The sensory encoder can be any function that maps observations to latent representations. For visual observations, the encoder may preserve spatial structure or collapse it into a global vector representation.

**Action encoder.** Each action $a_t \in \mathbb{R}^{n_a}$ is mapped to a latent action embedding via

$$\mathcal{E}_\psi^a : \mathbb{R}^{n_a} \to \mathbb{R}^{d_a}.$$

**Predictor.** The predictor $f_\theta : \mathbb{R}^{K \times d} \times \mathbb{R}^{K \times d_a} \to \mathbb{R}^d$ models transitions in the latent space. Given a history of $K$ past latent states and actions, it predicts the next latent state

$$\hat{z}_t = f_\theta \left( \{z_i\}_{i=t-K}^{t-1}, \{\mathcal{E}_\psi^a(a_i)\}_{i=t-K}^{t-1} \right). \tag{2}$$

### 3.2. Straightening latent trajectories

We seek to straighten the latent space induced by the sensory encoder $\mathcal{E}_\phi^s$ by penalizing the curvature along trajectories. Let $z_t, z_{t+1}$, and $z_{t+2}$ be three consecutive latent representations obtained by encoding observations $o_t, o_{t+1}$, and $o_{t+2}$ using $\mathcal{E}_\phi^s$. We define approximate latent velocity vectors

$$v_t = z_{t+1} - z_t, \quad v_{t+1} = z_{t+2} - z_{t+1}, \tag{3}$$

and seek to minimize the angle between them, or equivalently maximize their cosine similarity

$$\mathcal{C} = \frac{v_t \cdot v_{t+1}}{||v_t||_2 \cdot ||v_{t+1}||_2}. \tag{4}$$

### 3.3. Training objective.

The parameters $\phi, \psi$ and $\theta$ of the world model components $\mathcal{E}_\phi^s$, $\mathcal{E}_\psi^a$ and $f_\theta$ are trained jointly to minimize prediction error and enforce straightened trajectories.

**Prediction objective.** We minimize the MSE between the predicted and target latent states $\hat{z}_{t+1}$ and $z_{t+1}$:

$$\mathcal{L}_{pred} = ||\hat{z}_{t+1} - \text{sg}(z_{t+1})||_2^2, \tag{5}$$

where sg denotes the stop-gradient operation to prevent collapse of the latent space.

**Straightening objective.** We minimize trajectory curvatures by minimizing the negative cosine similarity

$$\mathcal{L}_{curv} = 1 - \mathcal{C}. \tag{6}$$

This straightening loss can be applied to any differentiable sensory encoder, either in isolation or jointly with the prediction objective.

**Overall objective.** The total training objective combines prediction and straightening as

$$\mathcal{L}_{total} = \mathcal{L}_{pred} + \lambda \mathcal{L}_{curv}, \tag{7}$$

where $\lambda \geq 0$ controls the strength of the straightening.

**Collapse prevention.** Since our encoder is trainable, the model is likely to produce degenerate solutions in which all latent representations collapse to a constant. Common anti-collapse strategies can be regularization-based (Bardes et al., 2022; Zhu et al., 2024; Balestriero & LeCun, 2025; Kuang et al., 2026), contrastive-based (Chen et al., 2020; He et al., 2020), and stop-gradient-based (Chen & He, 2021; Grill et al., 2020). Our curvature regularizer is *orthogonal to these anti-collapse methods* and can be combined with any of them. We use stop-grad for major experiments due to its simplicity and efficiency, as it does not require negative samples or introduce new hyperparameters. We apply stop-gradient to the target latent in the prediction loss (5) to prevent the gradients from $\mathcal{L}_{pred}$ from being backpropagated through the target branch. Although a collapsing solution is still possible in theory, stop-grad has been shown to be effective in self-supervised vision learning (Chen & He, 2021), and also effective in our experiments.

## 4. Planning with Straightened Dynamics

In this section, we present a theoretical analysis on the effect of straightening in the case of a linear dynamical system and show that straightened latent dynamics lead to better convergence in gradient-based planning.

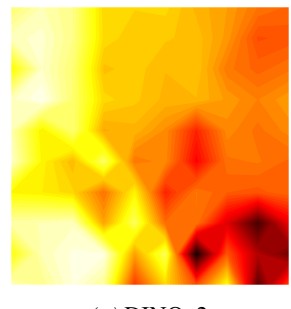 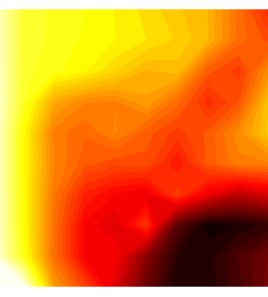

*(a)* DINOv2        *(b)* Straightened

*Figure 4.* Action-Space Loss Landscape. We pick one test sample from PushT with a planning horizon of 25 steps. For each coordinate $(a_x, a_y)$ in the grid, we fix the first action and optimize the remaining actions in the planning horizon to minimize the terminal goal cost. The heatmap represents the minimum attainable loss for each initial action choice, with darker colors indicating lower loss. The loss landscape is closer to convex after straightening.

We consider a goal-reaching task where we optimize an action sequence $\mathbf{a} = (a_0, \ldots, a_{K-1}) \in \mathbb{R}^{K \times d_a}$ over a horizon $K$ to reach a target latent goal $z_g$. For simplicity, we use the mean-squared terminal error,

$$\mathcal{L}(\mathbf{a}) = \|z_K - z_g\|_2^2, \qquad z_K = \Phi(\mathbf{a}), \tag{8}$$

where $\Phi$ denotes unrolling the learned latent dynamics from a fixed initial state $z_0$.

**Assumption 4.1** (Linear latent dynamics). For analysis, we consider linear latent dynamics $f$

$$f : (z_t, a_t) \to Az_t + Ba_t, \qquad \text{s.t.} \quad z_{t+1} = Az_t + Ba_t, \tag{9}$$

where $A \in \mathbb{R}^{d \times d}$ and $B \in \mathbb{R}^{d \times d_a}$. We first state results for $d_a = d$ and $B$ invertible; see Remark 4.5 for $d_a < d$.

**Definition 4.2** ($\varepsilon$-straight transition). Under the linear dynamics, we call $f$ $\varepsilon$-*straight* if

$$\|A - I\|_2 \leq \epsilon. \tag{10}$$

The term "straight" reflects that, as $\epsilon$ tends to 0, the dynamics of $f$ approach those of the reference function $g : (z_t, a_t) \to z_t + Ba_t$, where the state evolves linearly along a straight trajectory modified only by the control input. We are primarily interested in the regime where $\epsilon$ is small.

*Remark* 4.3 (Cosine similarity as a practical proxy). In practice, we regularize temporal straightness using the cosine similarity between consecutive latent velocities (4). Under mild bounded-variation assumptions on velocity magnitudes and smooth actions, a large cosine similarity implies that $(A - I)$ is small along visited velocity directions. Detailed proofs are in Section C.3.

**Theorem 4.4** (Conditioning of the planning Hessian). *Under Assumption 4.1 with $d_a = d$ and $B$ invertible, unrolling* (9) *yields*

$$z_K = A^K z_0 + \sum_{t=0}^{K-1} A^{K-1-t} Ba_t,$$

*so $z_K$ is affine in* **a** *and the planning Hessian is*

$$H := \nabla_{\mathbf{a}}^2 \mathcal{L}(\mathbf{a}) = 2J_\Phi^\top J_\Phi \succeq 0,$$

*where* $J_\Phi = \begin{bmatrix} A^{K-1}B, & A^{K-2}B, & \cdots, & B \end{bmatrix} \in \mathbb{R}^{d \times Kd}$. *Let* $\mathcal{W}_K := J_\Phi J_\Phi^\top = \sum_{k=0}^{K-1} A^k BB^\top (A^\top)^k$ *be the finite-horizon controllability Gramian (Kailath, 1980; Sontag, 1998; Chen, 1999). Then the* effective *condition number* $\kappa_{\text{eff}}(H) := \sigma_{\max}(H)/\sigma_{\min}^+(H)$ *satisfies*

$$\kappa_{\text{eff}}(H) = \kappa(\mathcal{W}_K) \le \kappa(B)^2 \frac{\sum_{k=0}^{K-1} \sigma_{\max}(A)^{2k}}{\sum_{k=0}^{K-1} \sigma_{\min}(A)^{2k}} \quad (11)$$
$$\le \kappa(B)^2 \, \kappa(A)^{2(K-1)}.$$

*where* $\kappa(A) := \sigma_{\max}(A)/\sigma_{\min}(A)$. *Moreover, if the transition is* $\varepsilon$*-straight with* $\varepsilon = \|A - I\|_2 < 1$, *then*

$$\kappa_{\text{eff}}(H) \le \kappa(B)^2 \left( \frac{1+\varepsilon}{1-\varepsilon} \right)^{2(K-1)}. \quad (12)$$

*For* $\varepsilon \le \frac{1}{2}$, *this gives* $\kappa_{\text{eff}}(H) \le \kappa(B)^2 e^{6\varepsilon K}$.

*Proofs are in Section C.2.*

*Remark* 4.5 (Low-dimensional actions). When $d_a < d$, $B$ is not invertible and $\mathcal{W}_K$ (hence $H$) may be singular outside the controllable subspace. Theorem 4.4 holds on $\mathcal{S}_K = \text{range}(\mathcal{W}_K)$ when $\kappa_{\text{eff}}$ is computed using $\sigma_{\min}^+(\mathcal{W}_K)$; see Section C.2.

Thm 4.4 shows that $\varepsilon$-straight transitions control the condition number of the planning Hessian: when $\varepsilon = \|A - I\|_2$ is small, the Gramian remains better conditioned, yielding $\kappa_{\text{eff}}(H)$ that grows slowly with the horizon. Since the planning objective is quadratic with Hessian $H \succeq 0$, gradient descent converges linearly at a rate controlled by the condition number, so the improved bounds on $\kappa_{\text{eff}}(H)$ translate to faster optimization in practice. For nonlinear predictors $z_{t+1} = f_\theta(z_t, a_t)$, analogous guarantees require controlling products of state-dependent Jacobians and higher-order terms, which can be an exciting future work direction.

Empirically, we observe that straightening yields a loss landscape with reduced non-convexity under nonlinear dynamics (Figure 4). In the next section, we show that it improves gradient-based planning.

# 5. Experiments

To test the effectiveness of the proposed method, we evaluate planning on four environments: Wall (Zhou et al., 2025; Sobal et al., 2025), PointMaze UMaze and a more complicated medium maze (Fu et al., 2020), and PushT (Chi et al., 2025). We compare against the baseline DINO-WM (Zhou et al., 2025) which builds on frozen DINOv2 spatial features or CLS tokens. Following DINO-WM's setup, we use a frameskip of 5 for all environments. Details on the environments and experiments are in Sections A and B.

## 5.1. Architecture details

Here, we describe the encoder and predictor architectures used to instantiate our world model.

**Visual encoder.** We consider two encoder setups for the visual encoder $\mathcal{E}_\phi^s$:

- A **frozen pretrained** visual backbone with a lightweight projector: We use DINOv2 (Oquab et al., 2024) as the backbone.[1] Given an observation, the backbone produces spatial features $e_t \in \mathbb{R}^{M \times D}$. We add a trainable lightweight CNN projector $\mathcal{P}_\phi$ on top of the backbone, leading to

$$z_t^v = \mathcal{P}_\phi(e_t) \in \mathbb{R}^{m_v \times d_v}, \quad (13)$$

where we usually choose $m_v \le M$ and $d_v \le D$. The projector may reduce spatial resolution (pooling/striding), channel dimension, or both, encouraging abstraction and reducing computation.

- A ResNet (He et al., 2015) trained **from scratch**, producing features $z_t^v \in \mathbb{R}^{m_v \times d_v}$ directly.

**Predictor.** We use a ViT (Dosovitskiy et al., 2021) as the dynamics predictor $f_\theta$. When available, proprioceptive states $p_t \in \mathbb{R}^{n_p}$ are encoded via $\mathcal{E}_\xi^p : \mathbb{R}^{n_p} \to \mathbb{R}^{d_p}$ and concatenated with each visual spatial feature. To condition on actions, we concatenate the action embeddings $z_t^a = \mathcal{E}_\psi^a(a_t) \in \mathbb{R}^{d_a}$ with the visual and proprioceptive embeddings before passing them to the predictor. We apply a temporal causal attention mask so tokens at time $t$ attend only to frames $\{t - K, \ldots, t - 1\}$, enabling frame-level autoregressive prediction.

**Cosine similarity computation.** The straightening loss (Eq. (4)) is only applied on visual latents $z_t^v$. Different implementations depend on whether latent representations preserve spatial structure:

- **Global features** ($n_v = 1$): Compute the cosine similarity directly between vectors.

- **Spatial features** ($n_v > 1$): We consider four variants: (i) compute the cosine similarity per-patch and average across patches; (ii) flatten all spatial features and then compute the cosine similarity; (iii) average-pool the spatial features before cosine similarity; (iv) use a learnable aggregation head to aggregate spatial features before cosine similarity. We use (iv) in the main experiments and ablate these choices in Section B.6.

---

[1]We choose DINOv2 because it has shown the best empirical performance for latent planning on 2D navigation tasks (Terver et al., 2025), outperforming DINOv3 (Siméoni et al., 2025) and V-JEPA2 (Assran et al., 2025).

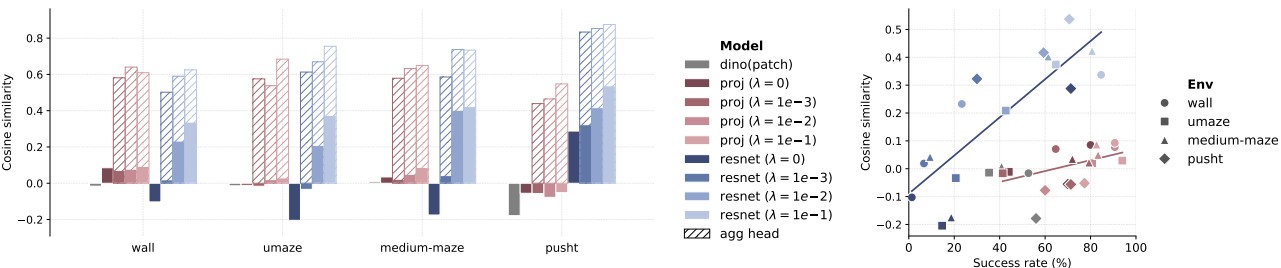

*Figure 5.* Latent Curvature and Open-Loop GD Success Rate for Different Encoders. Higher cosine similarity indicates lower curvature. Here, we compare models with spatial features and report the average patch-wise cosine similarity. Given the same type of encoder, reduced curvature generally leads to higher success rates.

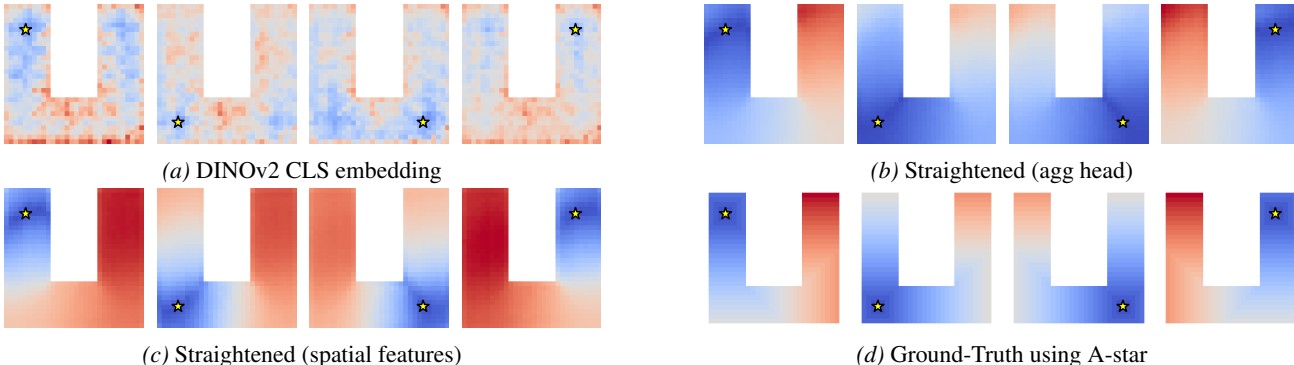

*(a)* DINOv2 CLS embedding *(b)* Straightened (agg head)

*(c)* Straightened (spatial features) *(d)* Ground-Truth using A-star

*Figure 6.* Distance heatmaps of PointMaze (blue indicates small values, and red indicates large values). The yellow star represents the target, and we compute the Euclidean distance between its embedding and those of all other states in the maze. Figures 6b and 6c use ResNet with output features $z \in \mathbb{R}^{14 \times 14 \times 8}$, trained with straightening regularization. After straightening, the latent distance accurately reflects the minimum number of steps required to reach the target.

## 5.2. How good is the embedding space?

We first inspect the learned embedding space before comparing the downstream planning performance. We measure latent trajectory curvatures and latent Euclidean distances to understand the impact of straightening. For interpretability, we train a VAE (Kingma & Welling, 2013; van den Oord et al., 2017) decoder with a reconstruction loss, detaching latents to stop gradients from the encoder and predictor.

We find that (i) *implicit straightening* can happen in JEPA world models when training the encoder using the prediction loss alone; (ii) adding explicit curvature regularization further strengthens and stabilizes the straightening effect; (iii) straightening encourages the latent Euclidean distance to better align with the geodesic distance; (iv) near-perfect reconstruction can be attained with a very low feature dimensionality.

**Reduced curvature.** In Figure 5, we compare the curvature of test latent trajectories by computing the cosine similarity of the difference in adjacent frames as in Equation (6). We also visualize the latent trajectories using PCA as shown in Figure 2 and Section E.2.

The pretrained DINOv2 embedding space is highly curved

as shown in the PCA plots and reflected by the low cosine similarities. The embedding space generally becomes straighter after training even without explicit straightening regularization. We attribute this *implicit straightening* to the JEPA objective: it favors representations whose temporal evolution is easy to predict, so training pressure reduces abrupt directional changes in the latent trajectory. With the *explicit straightening* regularization, the curvature of the embedding space is effectively reduced further. We observe that training a ResNet encoder from scratch generally yields lower curvatures than training a projector on top of a frozen pretrained backbone, likely because it offers greater representational flexibility to adapt the geometry to the dynamics.

When straightening is applied to the aggregation head, the curvature of the aggregated features is significantly reduced while the underlying spatial features are not forced to be overly straightened. For example, PushT has more complex object motions and the patch-wise cosine similarity is unable to faithfully capture the global state changes. The introduction of an aggregation head increases the flexibility of representation learning and generally leads to better planning performance (see Section B.6). We thus use this implementation for the main experiments.

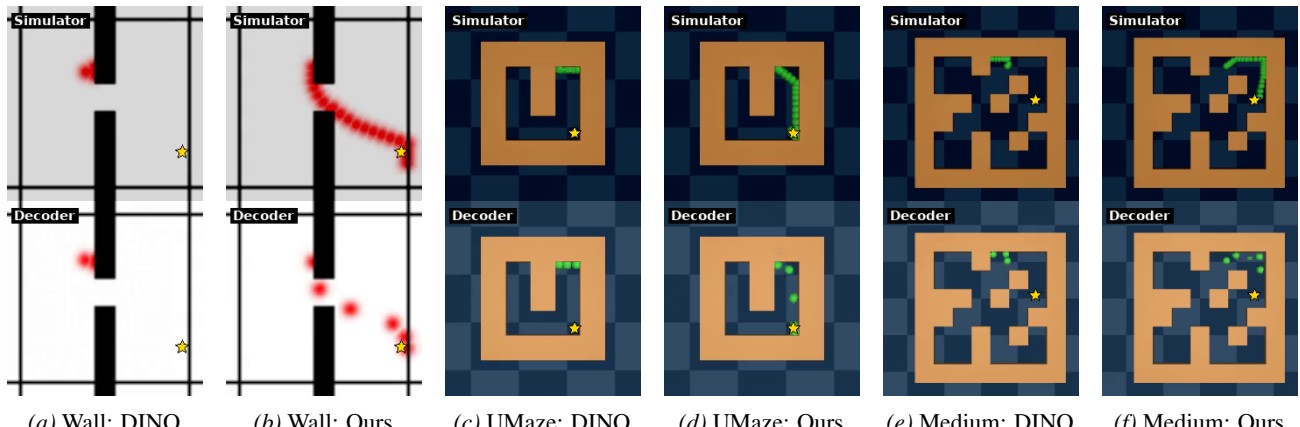

| *(a)* Wall: DINO | *(b)* Wall: Ours | *(c)* UMaze: DINO | *(d)* UMaze: Ours | *(e)* Medium: DINO | *(f)* Medium: Ours |

*Figure 7.* Comparison of Open-loop GD Planning. The star denotes the target. For each subfigure, the upper row shows the overlaid rendered images from the simulator by executing the actions, and the bottom shows imaginary rollouts (with a frameskip of 5) decoded using a trained decoder. GD planners are easily stuck with pretrained DINOv2 features, while straightening significantly increases success. More examples of open-loop planning are in Section E.3.

**Faithful distance.** Although DINOv2 is a strong visual encoder for various downstream vision tasks, it is not optimized for planning and control. As shown in Figure 2, MSE (which is equal to the squared Euclidean distance) between pretrained DINOv2 features does not reflect the progress of moving towards the target. To better understand the limitation of DINOv2, we visualize the Euclidean distance between the embedding of a target state and all other states in the maze in Figure 6. We also compare with ground-truth geodesic distance maps, computed using the A-star search algorithm on the grid of the maze. More heatmaps from different environments and encoders are in Section E.1.

Straightening results in a distance heatmap that closely aligns with the geodesic distance. Notably, the model is only trained on suboptimal, non-expert trajectories. Yet, it does not simply memorize the inefficient paths from the training data; instead, it learns to approximate the minimum number of steps required to transition between states. We also find that the spatial features and aggregated global features capture different levels of distance information. The spatial features preserve local geometry and thus yield fine-grained, locally discriminative distance variations, whereas global features provide a smoother, more coherent long-range signal that better reflects long-horizon distance-to-goal trends.

**Sufficient information.** To examine whether or not these projected features preserve sufficient information for planning, we train a decoder to reconstruct latents back to images. The decoder is solely for interpretability purposes and is detached from the world model via stop-gradient. Note that perfect reconstruction is not required, since planning only depends on task-relevant information. However, in our visually simple environments, even aggressively compressed features reconstruct the observations with high fidelity, as shown in Figure 7. This indicates that the resulting features retain sufficient planning-relevant information.

## 5.3. Planning

We show that straightening can significantly improve the planning success rates across models and environments.

**Setup.** The goals are sampled to guarantee they can be reached within 25 steps from the start states. We follow DINO-WM (Zhou et al., 2025) in using a frameskip of five, so we only need to rollout the world model for $H = 5$ times. During test time, an action sequence is optimized using gradient descent through the learned dynamics model ($f_\theta$) to minimize a goal cost. For PushT, we assume we have both target images and proprioceptions; for other environments, we only use target images to increase the task difficulty.

We evaluate performance in both open-loop and closed-loop settings. Open-loop planning optimizes a length-$H$ action sequence using the MSE between the terminal embedding and the target embedding as the planning cost. Closed-loop MPC replans at every step: it optimizes a length-$H$ action sequence, executes only the first action, and then replans, using a weighted objective that encourages the predicted trajectory to approach the target (see Section B.1). For PushT specifically, we use only the terminal loss within horizon $H$ because regime-switching dynamics make intermediate-state loss misleading, so we apply the weighted intermediate loss only beyond $H$ where it is more stable.

**Results.** As shown in Table 1, we observe a significant improvement across all models and environments. When training the projectors or encoders, we observe an improvement in performance even without the straightening regularization. We attribute this improvement to the *implicit straightening* during training as discussed in Section 5.2. For ResNet with spatial features, we observe abnormally low success rates for Wall, PointMaze-UMaze and PointMaze-Medium, which could be explained by the extremely high curvature

*Table 1.* Goal-reaching Success Rate of 50 Test Samples (%) using the GD planner. Values are mean ± std over three data sampling seeds. The best values are **bold**. The shaded rows are ours while the rest is DINO-WM (Zhou et al., 2025).

| Encoder | Config | | Wall | | PointMaze – UMaze | | PointMaze – Medium | | PushT | |
| --- | --- | --- | --- | --- | --- | --- | --- | --- | --- | --- |
| | dim | $\mathcal{L}_{curv}$ | Open-loop | MPC | Open-loop | MPC | Open-loop | MPC | Open-loop | MPC |
| DINOv2 (CLS) | $1 \times 384$ | ✗ | $28.67_{\pm 12.68}$ | $66.67_{\pm 10.50}$ | $25.33_{\pm 0.94}$ | $82.67_{\pm 9.98}$ | $20.00_{\pm 8.16}$ | $67.50_{\pm 3.54}$ | $19.33_{\pm 8.22}$ | $28.00_{\pm 1.63}$ |
| DINOv2 (patch) + proj | $1 \times 384$ | ✗ | $28.67_{\pm 0.94}$ | $76.00_{\pm 4.90}$ | $34.67_{\pm 1.89}$ | $79.33_{\pm 2.49}$ | $18.00_{\pm 1.63}$ | $46.00_{\pm 3.27}$ | $2.00_{\pm 1.63}$ | $11.33_{\pm 3.40}$ |
| DINOv2 (patch) + proj | $1 \times 384$ | ✓ | $42.00_{\pm 3.27}$ | $56.67_{\pm 4.11}$ | $38.67_{\pm 3.40}$ | $96.00_{\pm 0.00}$ | $22.67_{\pm 5.73}$ | $78.00_{\pm 2.83}$ | $5.33_{\pm 3.40}$ | $14.67_{\pm 0.94}$ |
| ResNet (from scratch) | $1 \times 384$ | ✗ | $4.67_{\pm 3.40}$ | $10.00_{\pm 1.63}$ | $82.00_{\pm 8.49}$ | $96.00_{\pm 1.63}$ | $66.00_{\pm 2.83}$ | $91.33_{\pm 0.94}$ | $2.00_{\pm 2.83}$ | $29.33_{\pm 3.40}$ |
| ResNet (from scratch) | $1 \times 384$ | ✓ | $84.00_{\pm 7.12}$ | $100.00_{\pm 0.00}$ | $52.00_{\pm 6.53}$ | $86.67_{\pm 0.94}$ | $54.00_{\pm 7.12}$ | $98.00_{\pm 0.00}$ | $19.33_{\pm 3.40}$ | $48.67_{\pm 4.99}$ |
| DINOv2 (patch) | $14 \times 14 \times 384$ | ✗ | $52.67_{\pm 5.73}$ | $76.67_{\pm 6.18}$ | $35.33_{\pm 4.11}$ | $80.67_{\pm 6.18}$ | $40.83_{\pm 10.07}$ | $76.67_{\pm 5.14}$ | $56.00_{\pm 4.32}$ | $66.00_{\pm 4.90}$ |
| DINOv2 (patch) + proj | $14 \times 14 \times 8$ | ✗ | $80.00_{\pm 7.12}$ | $90.67_{\pm 3.77}$ | $44.00_{\pm 7.12}$ | $81.33_{\pm 6.80}$ | $72.00_{\pm 4.32}$ | $96.67_{\pm 0.94}$ | $70.00_{\pm 1.63}$ | $78.67_{\pm 0.94}$ |
| DINOv2 (patch) + proj | $14 \times 14 \times 8$ | ✓ | $\mathbf{90.67}_{\pm 0.94}$ | $\mathbf{100.00}_{\pm 0.00}$ | $\mathbf{94.00}_{\pm 1.63}$ | $\mathbf{100.00}_{\pm 0.00}$ | $\mathbf{82.67}_{\pm 3.77}$ | $98.67_{\pm 0.94}$ | $\mathbf{77.33}_{\pm 6.18}$ | $85.33_{\pm 4.99}$ |
| ResNet (from scratch) | $14 \times 14 \times 8$ | ✗ | $1.33_{\pm 1.89}$ | $6.67_{\pm 1.89}$ | $14.67_{\pm 4.99}$ | $66.00_{\pm 9.09}$ | $18.67_{\pm 4.11}$ | $57.33_{\pm 4.71}$ | $71.33_{\pm 7.36}$ | $70.67_{\pm 10.50}$ |
| ResNet (from scratch) | $14 \times 14 \times 8$ | ✓ | $84.67_{\pm 2.49}$ | $\mathbf{100.00}_{\pm 0.00}$ | $64.67_{\pm 8.38}$ | $98.67_{\pm 1.89}$ | $80.67_{\pm 0.94}$ | $\mathbf{99.33}_{\pm 0.94}$ | $70.67_{\pm 0.94}$ | $\mathbf{91.33}_{\pm 2.49}$ |

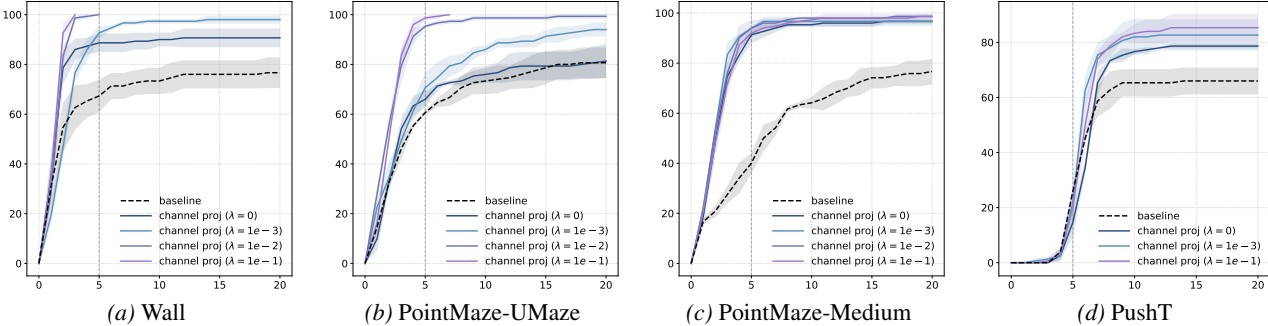

| *(a)* Wall | *(b)* PointMaze-UMaze | *(c)* PointMaze-Medium | *(d)* PushT |
| --- | --- | --- | --- |

*Figure 8.* Success Rate over MPC Steps. The dashed black lines represent DINO-WM with frozen DINOv2 patch features. The solid lines represent frozen DINOv2 patch features with a trainable channel projector (with resulting features $z \in \mathbb{R}^{14 \times 14 \times 8}$) with different strengths of straightening. Our model reaches 100% success rates very quickly as shown in Figures 8a and 8b.

in Figure 5, suggesting a degradation of features. We also notice that the implicit straightening is the weakest for the UMaze when using the projector, which also results in the lowest improvement in planning.

Applying explicit straightening further strengthens the straightness in the embedding space, resulting in more than 10% boost in open-loop and MPC success rates for almost all setups. For example, UMaze's open-loop success rate is improved from 44% to 94% with the projector, and 14.67% to 64.67% when training a ResNet from scratch. Note that we use weighted loss on intermediate states which enables reaching the target before consuming the full horizon $H = 5$. It is impressive that our model reaches 100% success with MPC on Wall and UMaze within only a few steps (Figure 8), suggesting it discovers more direct trajectories than the randomly generated test trajectories. The PushT success increases more slowly because we apply only the terminal loss within the horizon $H = 5$, yet straightening still yields substantial final gains. We also compare with other widely used temporal regularization, smoothness and temporal contrastiveness, in Section B.5, but find temporal straightening significantly more effective.

**Comparison of CEM and GD.** We compare the open-loop performance of gradient descent (GD) and the cross-entropy method (CEM) in Section B.3. Straightening regularization consistently improves the success rate of both

planners. For example, on Wall and PushT, it improves the projector baseline by roughly 10% for both GD and CEM. Overall, CEM achieves higher success rates but requires substantially longer planning time than GD. With straightening, GD achieves a better success–latency trade-off.

**Effect of feature dimensions.** We find that preserving spatial structure generally matters more than retaining channels. When we keep all patch tokens, we can aggressively reduce the channel dimension of DINOv2 features from 384 down to 8 without degrading performance. Increasing the channel dimension to $d \in \{32, 128\}$ does not improve performance and can even lead to a drop for some environments (Section B.4), which is not surprising as lower dimensions can simplify both dynamics prediction and downstream optimization. In contrast, collapsing patch features into a single global vector makes precise dynamics prediction harder. The predictor produces less accurate rollouts, which in turn reduces planning success. Notably, training a ResNet from scratch produces significantly better global features than training a global projector on frozen DINO patch features.

**Long horizon.** To further stress test our method, we also evaluate a longer-horizon setting where the target is 50 steps away. We leave out UMaze and Wall, because in those environments, a target picked via random 50-step rollouts can end up surprisingly close in terms of shortest-path distance, which does not reflect true long-horizon difficulty.

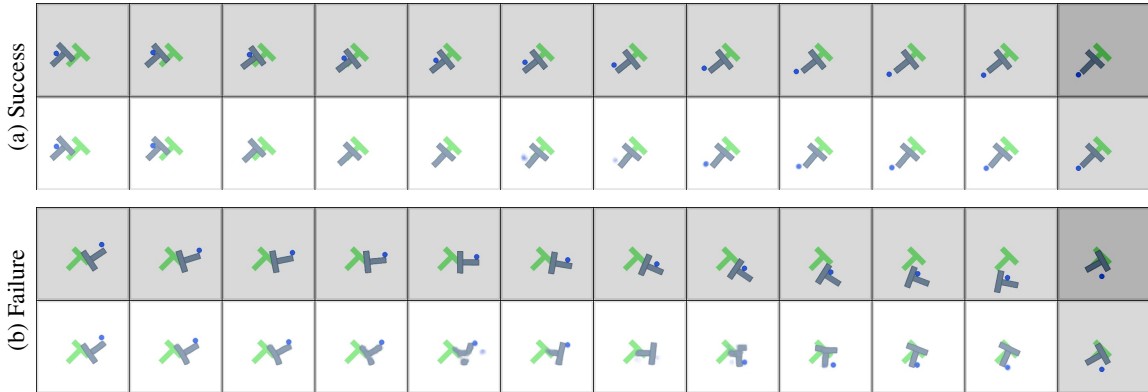

*Figure 9.* Examples of Long-Horizon Open-Loop GD Planning on PushT. For each example, the top row shows simulator-rendered images and the bottom row shows decoded images, with the last column being the target. The failure example shows a case where the long-horizon imagined rollout does not match the real dynamics.

We summarize the results in Table 2 and show success and failure examples in Figure 9. As expected, success rates drop substantially compared to the short-horizon setting, but our method consistently outperforms the baseline across all settings. More broadly, long-horizon rollouts remain a well-known challenge for latent planning where prediction errors compound over steps and lead to substantial trajectory drift. This is visible in failure cases where decoded rollouts become blurry or misaligned with the simulator.

Motivated by Figure 6, where the aggregation head produces a smoother long-range distance signal than spatial features alone, we add a global goal cost for long-horizon planning. Specifically, we keep the spatial goal cost and add a goal cost computed in the aggregated feature space: $\mathcal{L}_{\text{plan}} = \mathcal{L}_{\text{spatial}} + 0.1\mathcal{L}_{\text{agg}}$. Here, $\mathcal{L}_{\text{spatial}}$ measures squared goal distance over spatial features, while $\mathcal{L}_{\text{agg}}$ measures squared goal distance after applying the aggregation head. As shown in the last two rows of Table 2, this combined cost improves over using the spatial cost alone across all models under MPC. These results suggest that long-horizon planning may benefit from objectives that combine fine-grained local costs with global distance geometry.

**Teleported-PointMaze.** Pretrained visual embeddings primarily reflect visual similarity, whereas our straightening objective is designed to align the latent space with temporal dynamics. To test whether straightening truly captures dynamics rather than exploiting appearance cues, we introduce *Teleported-PointMaze* with modified transitions: touching the right wall instantly teleports the agent to the left side (see Section F). This creates states that look similar but have drastically different temporal distances, so relying on visual similarity alone can be misleading. We visualize a representative success case in Figure 27, where the straightened model plans to reach the target by leveraging teleportation.

**Limitations and future directions.** Our current formulation focuses on continuous goal-conditioned latent planning

*Table 2.* Longer-Horizon Success Rate (%) w/ Spatial Features.

| Model | $\mathcal{L}_{curv}$ | PushT | | PointMaze – Medium | |
|---|---|---|---|---|---|
| | | Open-loop | MPC | Open-loop | MPC |
| DINO-WM | – | $3.33 \pm _{2.36}$ | $27.33 \pm _{6.66}$ | $35.00 \pm _{2.35}$ | $65.33 \pm _{3.13}$ |
| + Proj | ✗ | $6.67 \pm _{3.77}$ | $26.67 \pm _{9.98}$ | $60.00 \pm _{3.27}$ | $72.00 \pm _{0.00}$ |
| + Proj | ✓ | $13.33 \pm _{3.77}$ | $24.00 \pm _{6.53}$ | $68.00 \pm _{8.64}$ | $88.00 \pm _{3.27}$ |
| ResNet | ✗ | $13.33 \pm _{3.77}$ | $29.33 \pm _{9.43}$ | $14.67 \pm _{6.80}$ | $48.00 \pm _{9.80}$ |
| ResNet | ✓ | $10.67 \pm _{4.99}$ | $33.33 \pm _{4.99}$ | $76.00 \pm _{6.53}$ | $98.67 \pm _{1.89}$ |
| *Combined planning cost:* $\mathcal{L}_{\text{plan}} = \mathcal{L}_{\text{spatial}} + 0.1\mathcal{L}_{\text{agg}}$ | | | | | |
| + Proj | ✓ | $20.00 \pm _{0.00}$ | $33.33 \pm _{4.16}$ | $66.67 \pm _{7.57}$ | $92.00 \pm _{5.29}$ |
| ResNet | ✓ | $13.33 \pm _{4.62}$ | $36.00 \pm _{5.29}$ | $68.67 \pm _{4.16}$ | $98.67 \pm _{1.15}$ |

with a symmetric Euclidean goal cost, which may be sub-optimal under asymmetric or irreversible dynamics. While our curvature regularizer straightens observed latent transitions and does not itself assume reversibility, such settings may require directional planning costs such as quasimetrics. Furthermore, the gains from using an aggregation head for long-horizon planning suggest that regularization and planning objectives do not necessarily operate in the prediction latent space: The world model can learn dynamics in one space, while the planner optimizes a task- and geometry-aware objective in a projected space.

## 6. Conclusion

In this work, we show that temporal straightening yields an embedding space that effectively facilitates latent planning. In this straightened representation space, the Euclidean distance provides a more reliable proxy for the geodesic distance and gradient-based planning is better conditioned. Across a range of 2D goal-reaching tasks, this leads to significant and consistent gains over baselines. More broadly, our findings highlight that representation geometry plays an important role in latent planning and show that straightening latent trajectories is a simple yet effective way to improve it. We believe this opens a promising path toward more efficient latent planning in more challenging environments.

## Impact Statement

This paper presents work whose goal is to advance the field of Machine Learning. World models with improved planning capabilities could have both beneficial applications (e.g., robotics, autonomous systems, scientific discovery) and potential risks if deployed without adequate safety measures. We encourage future work to consider safety implications when deploying such systems in real-world settings.

## Acknowledgments

We thank Yilun Kuang and Daohan Lu for helpful discussions. This work was supported in part by AFOSR under grant FA95502310139, NSF Award 1922658, Visko AI, a Google TPU Award, the NYU-KAIST Award A25-0081-002, and the Institute of Information & Communications Technology Planning Evaluation (IITP) under grant RS-2024-00469482, funded by the Ministry of Science and ICT (MSIT) of the Republic of Korea in connection with the Global AI Frontier Lab International Collaborative Research. The compute is supported by the NYU High Performance Computing resources, services, and staff expertise.

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

# Appendix

# A. Data and Environments

### A.1. Wall

This is a 2D navigation environment introduced by Zhou et al. (2025) and Sobal et al. (2025). The environment consists of two rooms separated by a wall with a single narrow door. To move between rooms, the agent must pass through this door. The task is to navigate from a start position to a target position, given start and target images. The action space consists of 2D vectors representing displacements in x and y axes. For training, we follow the approach of Zhou et al. (2025) to generate a dataset of 1,920 trajectories, each 50 time steps long. We train for 20 epochs.

### A.2. PointMaze (UMaze and Medium-Maze)

This is a 2D navigation environment based on the MuJoCo physics engine (Fu et al., 2020). We experiment on the "UMaze" and "Medium-Maze" here and plan to test other maze setups in future work. The task is to navigate from a start position to a target position, given start and target images. Unlike the previous "Wall" environment, this dynamics is governed by realistic physical properties such as velocity, acceleration, and inertia. The action space consists of forces applied along the x and y axes. For training, we follow Zhou et al. (2025) to generate a dataset of 2,000 trajectories for UMaze and 4,000 for Medium-Maze, each 100 time steps long. We train for 20 epochs.

### A.3. PushT

This is a challenging, contact-rich environment introduced by Chi et al. (2025). PushT features a pusher agent interacting with a T-shaped block. Starting from a random initial state, the agent must drive both the pusher and the T-block to a known feasible target configuration matching their target poses. The fixed green T is not the T-block's target and is only a visual reference marker. We use training data from Zhou et al. (2025), which contains 18500 trajectories with lengths of 100-300. We train for 2 epochs.

# B. Experiments

### B.1. Model Predictive Control (MPC)

We outline the MPC algorithm below. Unlike DINO-WM (Zhou et al., 2025) that uses Cross-Entropy Method (CEM) as the subplanner, we use gradient descent for the major experiments instead to accelerate planning.

a) **Encode States:** Given the current observation $o_0$ and the goal observation $o_g$ (both RGB images), we first encode them into their latent state representations using our trained encoder $\mathcal{E}^s$ (either a pre-trained DINOv2 encoder plus a projector, or a ResNet from scratch):

$$z_0 = \mathcal{E}^s(o_0), \quad z_g = \mathcal{E}^s(o_g).$$

b) **Initialize Actions:** An initial action sequence for the planning horizon $T$ is sampled from Gaussian distribution, $\{a_0, a_1, \ldots, a_{T-1}\}$.

c) **Define Objective:** The planning objective is to minimize the mean squared error (MSE) between the predicted final latent state $\hat{z}_T$ and the goal state $z_g$:

$$L = \|\hat{z}_T - z_g\|_2^2$$

where the latent trajectory is predicted by recursively applying the world model: $\hat{z}_t = f_\theta(\hat{z}_{t-1}, a_{t-1})$.

d) **Optimize via Gradient Descent:** Update actions iteratively using gradients of the cost with respect to the actions:

$$a_t \leftarrow a_t - \eta \frac{\partial L}{\partial a_t}, \quad \text{for } t = 0, \ldots, T-1,$$

where $\eta$ is the learning rate. Repeat until reaching the predefined number of iterations.

e) **Execute Action:** After the optimization loop is complete, the first $k$ actions from the optimized action sequence are executed in the environment.

f) **Re-plan:** The process is repeated from step (a) at the next environment timestep, using the new observation $o_1$.

## B.2. Hyperparameters

*Table 3.* Training Hyperparameters.

| Name | Value |
|------|-------|
| Projector/ResNet lr | 1e-5[a] |
| Predictor lr | 5e-4 |
| Action/Prop encoder lr | 5e-4 |
| Batch size | 32 |
| History frames | 3 |
| Frameskip | 5 |

[a]We observe severe performance degradation when training without straightening and decreasing the learning rate helps. We thus use $lr = 1e - 6$ for no straightening.

*Table 4.* Planning Hyperparameters.

| Name | Value |
|------|-------|
| Subplanner horizon | 25 |
| # Executed actions | 25[a] |
| Optimizer | Adam |
| Action Initialization | Zero |
| Learning rate | 0.1 |
| #opt steps | 100 |

[a]This is for open-loop. If using MPC, we execute the first 5 actions (or the first chunk of actions if using a frameskip of 5).

## B.3. Planning: GD v.s. CEM

We compare the open-loop success rate using GD and CEM planners. For one plan, GD optimizes the action sequence by backpropagating through the learned rollout model. With $N$ optimization steps, this requires $N$ forward rollouts and $N$ backward passes. In contrast, CEM iteratively samples $M$ candidate action sequences, rolls each candidate out with the learned predictor, refits the sampling distribution to the top-performing candidates, and repeats this procedure for $K$ iterations. Thus, CEM requires $MK$ forward rollouts, with $M$ typically large for competitive performance.

In our experiments, we find that CEM requires at least 200 samples and 10 iterations to achieve strong performance, making it roughly $10\times$ slower than GD in wall-clock planning time. We report wall-clock time for open-loop planning over 50 test trajectories on a single L40S GPU in Figure 10.

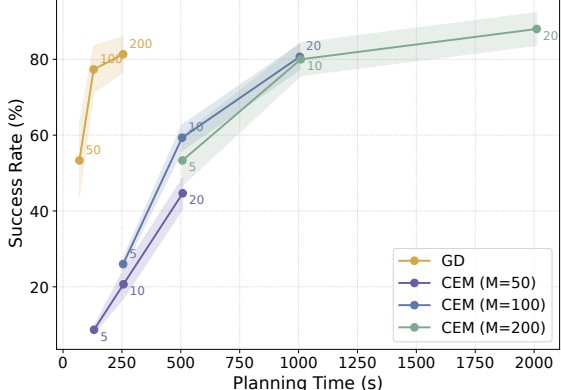

*Figure 10.* Success rate versus wall-clock planning time for open-loop GD and CEM planning after straightening.

As shown in Table 5, straightening consistently improves **both** GD and CEM across environments and model architectures. Consistent with prior work (Zhou et al., 2025), CEM often obtains higher absolute success rates than GD, but at substantially higher computational cost. Importantly, straightening largely reduces the performance gap between GD and CEM, suggesting that the regularizer improves the latent optimization landscape and enables simple gradient-based planning to achieve a better success–latency trade-off.

*Table 5.* Goal-reaching Success Rate of 50 Test Samples (%) in open-loop planning. We compare GD and CEM planners. Values are mean ± std over three data seeds. The best value is **bold**.

| Method | Config | | Wall | | PointMaze – Umaze | | PointMaze – Medium | | PushT | |
|--------|--------|--------------|------|------|-------------------|------|--------------------|------|-------|------|
| | dim | $\mathcal{L}_{curv}$ | GD | CEM | GD | CEM | GD | CEM | GD | CEM |
| DINOv2 (patch) | $14 \times 14 \times 384$ | ✗ | $73.33 \pm 3.40$ | $87.33 \pm 4.99$ | $63.33 \pm 8.22$ | $88.00 \pm 1.63$ | $70.00 \pm 4.08$ | $88.00 \pm 1.63$ | $62.67 \pm 4.11$ | $71.33 \pm 7.72$ |
| DINOv2 (patch) + proj | $14 \times 14 \times 8$ | ✗ | $80.00 \pm 7.12$ | $92.00 \pm 0.00$ | $44.00 \pm 7.12$ | $75.33 \pm 4.99$ | $72.00 \pm 4.32$ | $\mathbf{92.67} \pm 4.71$ | $70.00 \pm 1.63$ | $71.33 \pm 6.18$ |
| DINOv2 (patch) + proj | $14 \times 14 \times 8$ | ✓ | $\mathbf{90.67} \pm 0.94$ | $\mathbf{100.00} \pm 0.00$ | $\mathbf{94.00} \pm 1.63$ | $\mathbf{94.00} \pm 1.63$ | $\mathbf{82.67} \pm 3.77$ | $86.67 \pm 1.89$ | $\mathbf{77.33} \pm 6.18$ | $\mathbf{80.00} \pm 4.32$ |
| ResNet | $14 \times 14 \times 8$ | ✗ | $1.33 \pm 1.89$ | $1.33 \pm 0.94$ | $14.67 \pm 4.99$ | $20.67 \pm 0.94$ | $18.67 \pm 4.11$ | $24.00 \pm 4.32$ | $71.33 \pm 7.36$ | $56.00 \pm 0.00$ |
| ResNet | $14 \times 14 \times 8$ | ✓ | $84.67 \pm 2.49$ | $90.00 \pm 5.89$ | $64.67 \pm 8.38$ | $83.33 \pm 2.49$ | $80.67 \pm 0.94$ | $89.33 \pm 6.18$ | $70.67 \pm 0.94$ | $72.67 \pm 6.60$ |

## B.4. Effect of Feature Dimensions

In order to improve efficiency and efficacy, we ablate the output dimensions of the encoders. Here, we test on the "frozen DINOv2 + spatial projector" setup and preserve the spatial dimensions of the DINOv2 patch features $m_v = 196$ but decreasing channels from 384 to $d_v \in \{2, 8, 32, 128\}$. For all experiments, we use $lr = 1e - 6$ for the encoder. If with straightening, we apply straightening on the aggregation head as described in Section B.6 with a straightening strength $\lambda = 0.1$.

We report the open-loop planning success rate of 50 test samples over three data sampling seeds in Figure 11. Very small dimensions (e.g., $d_v = 2$) result in poor performance, indicating insufficient capacity to preserve planning-relevant information. Increasing to a moderate dimension ($d_v = \{8, 32\}$) yields the best results, while too large dimensions ($d_v = 128$) consistently reduce success rates. This suggests that overly high-dimensional latents can hinder gradient-based planning.

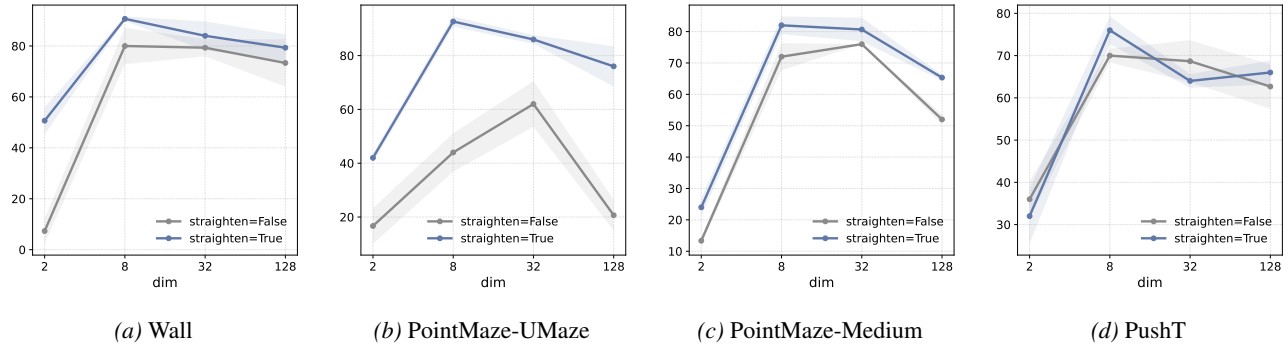

*(a)* Wall     *(b)* PointMaze-UMaze     *(c)* PointMaze-Medium     *(d)* PushT

*Figure 11.* Comparison of Different Dimensions. The line plots show the success rate changes with increasing channels. Too small dimensions (e.g. $d_v = 2$) are unable to encode sufficient planning-relevant information, while unnecessarily high dimensions (e.g. $d_v = 128$) hinder the planning performance.

## B.5. Comparison to Smoothness and Temporal Contrastive Objectives

We compare with two common temporal regularization objectives:

- **Smoothness.** This objective penalizes large temporal jumps in visual embeddings:

$$\mathcal{L}_{\text{smooth}} = \mathbb{E}_t \left[ \|z_{t+1} - z_t\|_2^2 \right].$$

  However, an overly strong smoothness penalty can lead to degenerate solutions where embeddings of different states collapse to similar values.

- **Time contrastiveness.** This objective treats frames within a temporal window of size $k$ as positives and other frames in the same trajectory as negatives, encouraging temporally nearby embeddings to be similar and temporally distant embeddings to be different:

$$\mathcal{L}_{\text{tc}} = \text{InfoNCE}(z_{\text{pos}}, z_{\text{neg}}).$$

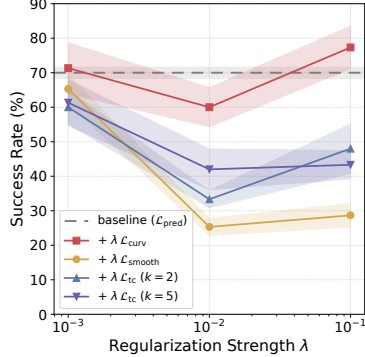

*Figure 12.* Comparison with Other Regularizations.

However, when training trajectories are suboptimal, temporal distance may not reflect geodesic distance: states that are geodesically close can be temporally far apart, and this objective may incorrectly separate them.

We test on the "frozen DINOv2 + spatial projector" setup and report open-loop GD planning success rate on PushT in Figure 12, evaluated on 50 test samples over three data seeds. Overall, we do not observe improvements from adding smoothness or temporal contrastive objectives. Larger weights generally hurt performance, while smaller weights are less harmful but still do not match the gains from straightening. These objectives may still be useful in other settings, and our method is complementary to them: the curvature regularization loss can be combined with any other losses.

## B.6. Cosine Similarity Variants for Spatial Features

For spatial visual features $z_t^v \in \mathbb{R}^{m_v \times d_v}$ ($m_v > 1$), we compute straightness from approximate latent velocities $v_t := z_{t+1}^v - z_t^v \in \mathbb{R}^{m_v \times d_v}$. Let $v_{t,i} \in \mathbb{R}^{d_v}$ denote the $i$-th patch vector and $\cos(u, w) = \frac{u^\top w}{\|u\|_2 \|w\|_2}$. We ablate four choices of $\mathcal{C}_t$:

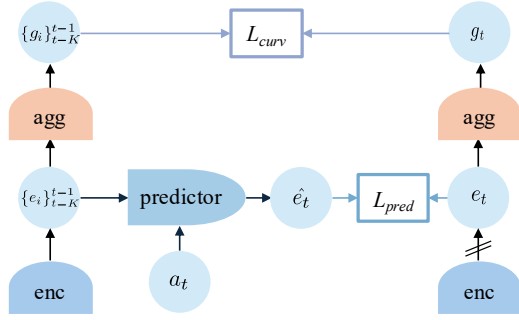

- **[patch]** We treat each patch independently, then average:

$$\mathcal{C}_t = \frac{1}{m_v} \sum_{i=1}^{m_v} \cos(v_{t,i}, v_{t+1,i}).$$

- **[mean]** We average patches to one vector, then cosine:

$$\bar{v}_t = \frac{1}{m_v} \sum_{i=1}^{m_v} v_{t,i}, \qquad \mathcal{C}_t = \cos(\bar{v}_t, \bar{v}_{t+1}).$$

*Figure 13.* Aggregation Head for Straightening during Training. The prediction loss is applied to spatial features, while the curvature loss is applied to the aggregated features.

- **[flatten]** We flatten the spatial features and single cosine over all dimensions:

$$\mathcal{C}_t = \cos(\text{vec}(v_t), \text{vec}(v_{t+1})),$$

where $\text{vec}(\cdot) : \mathbb{R}^{m_v \times d_v} \to \mathbb{R}^{m_v d_v}$.

- **[agg]** We learn an aggregation head to aggregate features to a single global feature before cosine (Figure 13):

$$\mathcal{C}_t = \cos(h_\phi(v_t), h_\phi(v_{t+1})),$$

with an aggregation head $h_\phi : \mathbb{R}^{m_v \times d_v} \to \mathbb{R}^{d_h}$. Concretely, we use an MLP with an output dimension of 128 as $h_\phi$ in all experiments.

We test these variants on the "frozen DINOv2 + spatial projector" setup and report the open-loop planning success rate of 50 test samples over three data sampling seeds in Figure 14. The projector projects pretrained DINOv2 patch features $e_t \in \mathbb{R}^{196 \times 384}$ to $z_t \in \mathbb{R}^{196 \times 8}$. For the straightening strength coefficient, we use $\lambda = 0.1$ for agg and $\lambda = 0.01$ for the rest, as these values yield the best performance. We find that using a learnable aggregation head performs best. This is not surprising as straightening should act on the *global* trajectory representations, whereas spatial tokens mainly capture local, patch-level variations that are only loosely aligned across time due to object motion and occlusion.

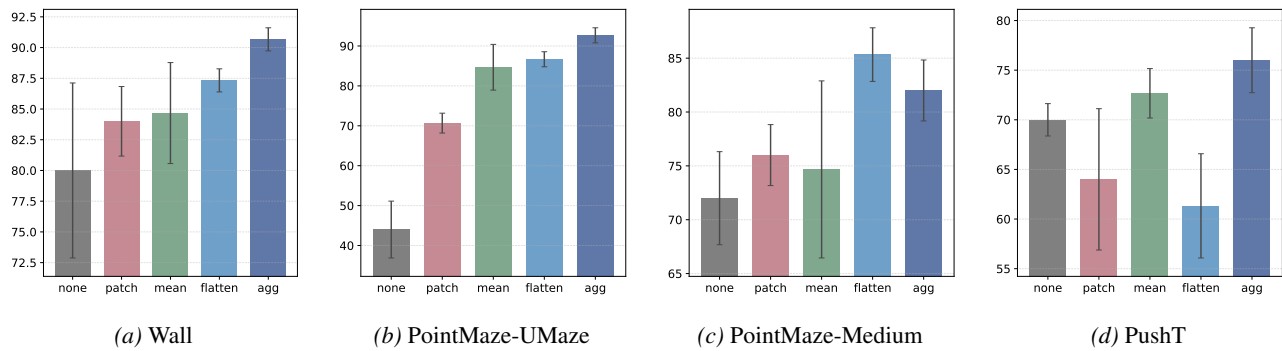

| *(a)* Wall | *(b)* PointMaze-UMaze | *(c)* PointMaze-Medium | *(d)* PushT |

*Figure 14.* Comparison of Different Straightening Strategies. The bar charts show the planning success rates. While all cosine similarity variants lead to better performance than no straightening, adding a learnable aggregation head gives the best performance.

# C. Theoretical Analysis

## C.1. Setup and notation

We optimize an action sequence $\mathbf{a} = (a_0, \ldots, a_{K-1}) \in \mathbb{R}^{K \times d_a}$ over horizon $K$ to minimize the terminal MSE

$$\mathcal{L}(\mathbf{a}) = \|z_K - z_g\|_2^2, \qquad z_K = \Phi(\mathbf{a}), \tag{14}$$

where $\Phi$ denotes unrolling the latent dynamics from a fixed initial state $z_0$.

**Assumption C.1** (Linear latent dynamics). We assume linear latent dynamics

$$z_{t+1} = Az_t + Ba_t, \qquad A \in \mathbb{R}^{d \times d}, \ B \in \mathbb{R}^{d \times d_a}. \tag{15}$$

**Definition C.2** (Effective condition number). For a PSD matrix $H \succeq 0$ with a nontrivial nullspace, define

$$\kappa_{\text{eff}}(H) := \frac{\sigma_{\max}(H)}{\sigma_{\min}^+(H)},$$

where $\sigma_{\min}^+(H)$ is the smallest nonzero singular value.

**Definition C.3** ($\varepsilon$-straight transition). In the linear model (15), define

$$\varepsilon := \|A - I\|_2.$$

## C.2. Conditioning of the planning Hessian

Unrolling (15) gives the affine terminal map

$$z_K = A^K z_0 + \sum_{t=0}^{K-1} A^{K-1-t} B a_t. \tag{16}$$

Define the rollout Jacobian

$$J_\Phi := \frac{\partial z_K}{\partial \mathbf{a}} = \begin{bmatrix} A^{K-1}B, & A^{K-2}B, & \cdots, & B \end{bmatrix} \in \mathbb{R}^{d \times (Kd_a)}. \tag{17}$$

The associated finite-horizon discrete controllability Gramian is

$$\mathcal{W}_K := J_\Phi J_\Phi^\top = \sum_{k=0}^{K-1} A^k B B^\top (A^\top)^k \in \mathbb{R}^{d \times d}, \tag{18}$$

a standard term in linear systems (Kailath, 1980; Sontag, 1998; Chen, 1999).

**Lemma C.4** (Hessian form and Gramian equivalence). *Under* (14)–(15), *the planning Hessian satisfies*

$$H := \nabla_{\mathbf{a}}^2 \mathcal{L}(\mathbf{a}) = 2J_\Phi^\top J_\Phi \succeq 0. \tag{19}$$

*Moreover, the nonzero singular values of $J_\Phi^\top J_\Phi$ equal those of $J_\Phi J_\Phi^\top$, hence*

$$\kappa_{\text{eff}}(H) = \kappa(\mathcal{W}_K). \tag{20}$$

*Proof.* $H$ is positive semi-definite by definition. Since $z_K$ is affine in $\mathbf{a}$ by (16), $\mathcal{L}(\mathbf{a}) = \|z_K - z_g\|_2^2$ is a convex quadratic, and direct differentiation yields $H = 2J_\Phi^\top J_\Phi \succeq 0$. For any matrix $M$, the nonzero eigenvalues of $M^\top M$ and $MM^\top$ coincide. Applying this with $M = J_\Phi$ gives that the nonzero eigenvalues of $H/2$ equal those of $\mathcal{W}_K$, which implies (20). $\square$

**Theorem C.5** (Conditioning bound). *Assume* (15). *Consider first the square-action case $d_a = d$ with $B$ invertible. Then*

$$\kappa_{\text{eff}}(H) = \kappa(\mathcal{W}_K) \ \leq \ \kappa(B)^2 \frac{\sum_{k=0}^{K-1} \sigma_{\max}(A)^{2k}}{\sum_{k=0}^{K-1} \sigma_{\min}(A)^{2k}} \ \leq \ \kappa(B)^2 \kappa(A)^{2(K-1)}, \tag{21}$$

*where $\kappa(A) := \sigma_{\max}(A)/\sigma_{\min}(A)$. If additionally $\varepsilon = \|A - I\|_2 < 1$, then*

$$\kappa_{\text{eff}}(H) \ \leq \ \kappa(B)^2 \left( \frac{1+\varepsilon}{1-\varepsilon} \right)^{2(K-1)} \ \leq \ \kappa(B)^2 e^{6\varepsilon K} \quad (\varepsilon \leq \tfrac{1}{2}). \tag{22}$$

*Proof.* By Lemma C.4, it suffices to bound $\kappa(\mathcal{W}_K)$.

**Upper bound.** For any unit vector $x \in \mathbb{R}^d$,

$$x^\top \mathcal{W}_K x = \sum_{k=0}^{K-1} \|B^\top (A^\top)^k x\|_2^2 \le \sum_{k=0}^{K-1} \|B\|_2^2 \|A^k\|_2^2 \|x\|_2^2 \le \sigma_{\max}(B)^2 \sum_{k=0}^{K-1} \sigma_{\max}(A)^{2k}.$$

Taking the maximum over $\|x\|_2 = 1$ yields

$$\lambda_{\max}(\mathcal{W}_K) \le \sigma_{\max}(B)^2 \sum_{k=0}^{K-1} \sigma_{\max}(A)^{2k}.$$

**Lower bound.** Since $B$ is invertible, $\|B^\top u\|_2 \ge \sigma_{\min}(B)\|u\|_2$ for all $u$. Also $\sigma_{\min}(A^k) \ge \sigma_{\min}(A)^k$. Thus for any unit $x$,

$$\|B^\top (A^\top)^k x\|_2 \ge \sigma_{\min}(B)\,\|(A^\top)^k x\|_2 \ge \sigma_{\min}(B)\,\sigma_{\min}(A^k)\,\|x\|_2 \ge \sigma_{\min}(B)\,\sigma_{\min}(A)^k,$$

hence

$$x^\top \mathcal{W}_K x \ge \sigma_{\min}(B)^2 \sum_{k=0}^{K-1} \sigma_{\min}(A)^{2k}.$$

Taking the minimum over $\|x\|_2 = 1$ yields

$$\lambda_{\min}(\mathcal{W}_K) \ge \sigma_{\min}(B)^2 \sum_{k=0}^{K-1} \sigma_{\min}(A)^{2k}.$$

**Combine.** Dividing the two bounds gives the first inequality in (21). For the second, use positivity of terms:

$$\frac{\sum_{k=0}^{K-1} \sigma_{\max}(A)^{2k}}{\sum_{k=0}^{K-1} \sigma_{\min}(A)^{2k}} \le \max_{0 \le k \le K-1} \frac{\sigma_{\max}(A)^{2k}}{\sigma_{\min}(A)^{2k}} = \kappa(A)^{2(K-1)}.$$

$\varepsilon$**-specialization.** If $\varepsilon = \|A - I\|_2 < 1$, then by Weyl's perturbation theorem, $\sigma_{\max}(A) \le 1 + \varepsilon$ and $\sigma_{\min}(A) \ge 1 - \varepsilon$, which implies the first inequality in (22). For $\varepsilon \le \frac{1}{2}$, the standard bound $\ln\!\left(\frac{1+\varepsilon}{1-\varepsilon}\right) \le 3\varepsilon$ gives the exponential form. $\qquad\square$

*Remark* C.6 (Low-dimensional actions $d_a < d$). If $d_a < d$, then $B$ is not invertible and $\mathcal{W}_K$ may be singular. All statements hold on the controllable subspace $\mathcal{S}_K = \mathrm{range}(\mathcal{W}_K)$ by replacing $\lambda_{\min}(\mathcal{W}_K)$ with $\lambda_{\min}^+(\mathcal{W}_K)$ and interpreting $\kappa(\mathcal{W}_K)$ as an effective condition number. In this case, additional controllability assumptions are needed to lower bound $\sigma_{\min}^+(\mathcal{W}_K)$.

### C.3. Cosine similarity as a proxy

**Assumption C.7** (Constant velocity and smooth actions). Define latent velocities $v_t := z_{t+1} - z_t$. Assume there exists a constant $c > 0$ such that

$$\|v_t\|_2 = c \qquad \text{for all } t = 0, \ldots, K-1.$$

Assume action smoothness $\Delta_a := \max_t \|a_{t+1} - a_t\|_2 < \infty$.

**Definition C.8** (Cosine similarity). For $t = 0, \ldots, K-2$, define

$$\mathcal{C}_t := \cos(v_t, v_{t+1}) = \frac{v_t^\top v_{t+1}}{\|v_t\|_2 \|v_{t+1}\|_2}, \qquad \bar{\mathcal{C}} := \frac{1}{K-1} \sum_{t=0}^{K-2} \mathcal{C}_t.$$

**Proposition C.9** (Cosine proxy $\Rightarrow$ small $(A - I)$ along visited directions). *Under linear dynamics* (15), *let* $\hat{v}_t := v_t / \|v_t\|_2$. *Under Assumption C.7, for each* $t = 0, \ldots, K-2$,

$$\|(A - I)\hat{v}_t\|_2 \le \sqrt{2(1 - \mathcal{C}_t)} + \frac{\sigma_{\max}(B)\Delta_a}{c}. \tag{23}$$

*If* $\bar{\mathcal{C}} \ge 1 - \eta$, *then*

$$\frac{1}{K-1} \sum_{t=0}^{K-2} \|(A - I)\hat{v}_t\|_2 \le \sqrt{2\eta} + \frac{\sigma_{\max}(B)\Delta_a}{c}. \tag{24}$$

*Proof.* Under (15),

$$v_{t+1} - v_t = (z_{t+2} - z_{t+1}) - (z_{t+1} - z_t) = (A - I)(z_{t+1} - z_t) + B(a_{t+1} - a_t) = (A - I)v_t + B(a_{t+1} - a_t).$$

Thus, by the triangle inequality,

$$\|(A - I)\hat{v}_t\|_2 = \frac{\|(A - I)v_t\|_2}{\|v_t\|_2} \leq \frac{\|v_{t+1} - v_t\|_2}{\|v_t\|_2} + \frac{\|B(a_{t+1} - a_t)\|_2}{\|v_t\|_2} \leq \frac{\|v_{t+1} - v_t\|_2}{c} + \frac{\sigma_{\max}(B)\Delta_a}{c}.$$

Since $\|v_t\|_2 = \|v_{t+1}\|_2 = c$,

$$\|v_{t+1} - v_t\|_2^2 = \|v_{t+1}\|_2^2 + \|v_t\|_2^2 - 2v_{t+1}^\top v_t = 2c^2(1 - \mathcal{C}_t),$$

hence $\|v_{t+1} - v_t\|_2/c = \sqrt{2(1 - \mathcal{C}_t)}$, proving (23). Averaging and applying Jensen's inequality to the concave map $x \mapsto \sqrt{x}$ gives

$$\frac{1}{K-1} \sum_{t=0}^{K-2} \sqrt{1 - \mathcal{C}_t} \leq \sqrt{1 - \bar{\mathcal{C}}} \leq \sqrt{\eta},$$

which implies (24). $\qquad\square$

*Remark* C.10 (Directional vs. spectral control). Proposition C.9 bounds $(A-I)$ only along visited directions $\{\hat{v}_t\}$. Upgrading this to a uniform spectral bound $\varepsilon = \|A - I\|_2$ requires an additional coverage condition so that visited directions span the latent space. This is not an impractical assumption since training trajectories are typically collected to be diverse. Under such regimes, maximizing cosine similarity provides a meaningful proxy for making $A$ close to $I$ in spectral norm.

# D. Related Work (Cont.)

Here, we discuss the connections and differences between temporal straightening and local linearization, Koopman methods, and the broader literature on learning plannable representations. Temporal straightening targets the curvature of latent trajectories, a geometric property distinct from linear dynamics, whether local or global.

Local model-based control approximates nonlinear dynamics around a nominal trajectory using first- or second-order models, as in DDP and iLQR (Mayne, 1966; Li & Todorov, 2004). Because these methods presuppose a low-dimensional state space, they have motivated representation-learning methods that map high-dimensional observations into latent spaces where local dynamics models become applicable. For example, E2C and RCE explicitly impose locally linear latent dynamics (Watter et al., 2015; Banijamali et al., 2018), while later methods broaden this direction by learning representations or objectives that support locally linear control (Zhang et al., 2019a; Levine et al., 2020; Shu et al., 2020). Temporal straightening differs in both goal and mechanism. We do not aim to learn locally linear dynamics, nor do we design representations for locally linear control. Instead, we jointly learn the encoder and predictor while directly regularizing the geometry of latent trajectories.

Koopman methods seek observables whose evolution is linear under a global operator (Koopman, 1931). Recent deep models learn such observables or eigenfunctions directly from data (Lusch et al., 2018; Yeung et al., 2019; Takeishi et al., 2017; Cheng et al., 2026). Temporal straightening targets a different property of the learned representation. Koopman methods are more restrictive in the dynamics model but do not require latent trajectories to be straight as linear systems can still produce curved or oscillatory paths. In contrast, temporal straightening allows nonlinear latent dynamics through a learned predictor, but more directly constrains trajectory geometry by penalizing curvature along observed transitions.

More broadly, our work is related to learning plannable representations, especially methods that make planning easier by aligning latent geometry or value structure with feasible transitions. This includes approaches where planning or inference can be carried out through interpolation in latent space (Eysenbach et al., 2024; Kurutach et al., 2018; Wang et al., 2019), as well as methods that build planning-oriented geometry from shortest-path, reachability, or asymmetric goal-reaching structure (Yang et al., 2020; Wang et al., 2023). These works share the premise that representation geometry or value structure matters for planning, but with different focuses. Our contribution is to use straightening itself as a simple geometry regularization during training.

# E. Visualizations

## E.1. Distance Heatmaps

We plot heatmaps of the Euclidean distances in the embedding space. The yellow star represents the target, and we compute the Euclidean distance between its embedding and those of all other states in the maze. Blue indicates small values, and red indicates large values. With straightening, the latent distance accurately reflects the minimum number of steps required to reach the target. We find that spatial and global/aggregated features capture complementary distance information: spatial features preserve local geometry and yield fine-grained, locally discriminative distances, while global/aggregated features provide an informative longer-range signal.

To compare the distance heatmaps, we compare with ground-truth heatmaps constructed by dividing the mazes into discrete grids and applying the A-star algorithm. 4-neighbor connectivity means each grid cell connects only to up/down/left/right cells. 8-neighbor connectivity adds the four diagonals (up-left, up-right, down-left, down-right), so paths can cut corners diagonally and distances are usually shorter.

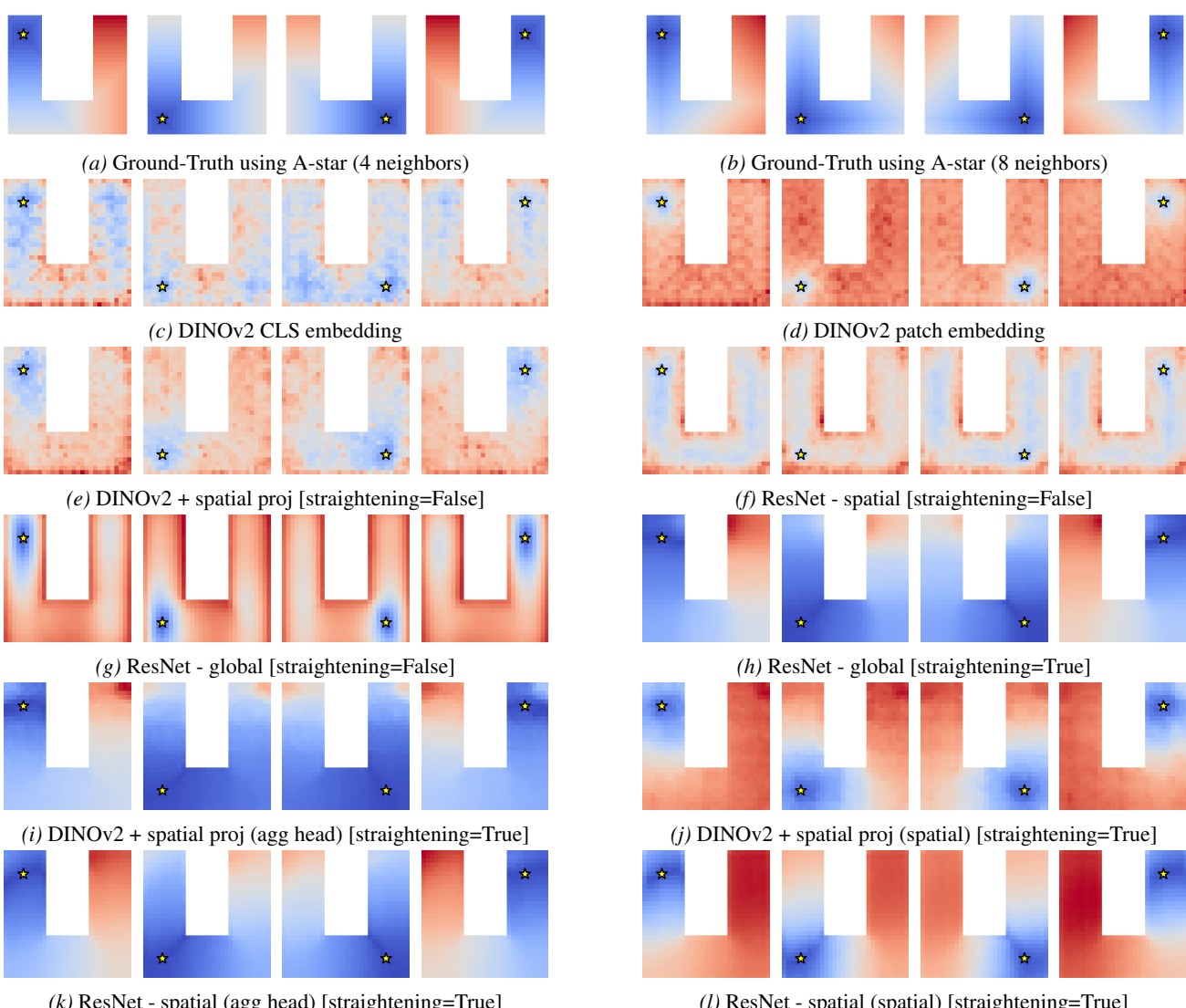

*(a)* Ground-Truth using A-star (4 neighbors)  *(b)* Ground-Truth using A-star (8 neighbors)

*(c)* DINOv2 CLS embedding  *(d)* DINOv2 patch embedding

*(e)* DINOv2 + spatial proj [straightening=False]  *(f)* ResNet - spatial [straightening=False]

*(g)* ResNet - global [straightening=False]  *(h)* ResNet - global [straightening=True]

*(i)* DINOv2 + spatial proj (agg head) [straightening=True]  *(j)* DINOv2 + spatial proj (spatial) [straightening=True]

*(k)* ResNet - spatial (agg head) [straightening=True]  *(l)* ResNet - spatial (spatial) [straightening=True]

*Figure 15.* Distance heatmaps of PointMaze-UMaze.

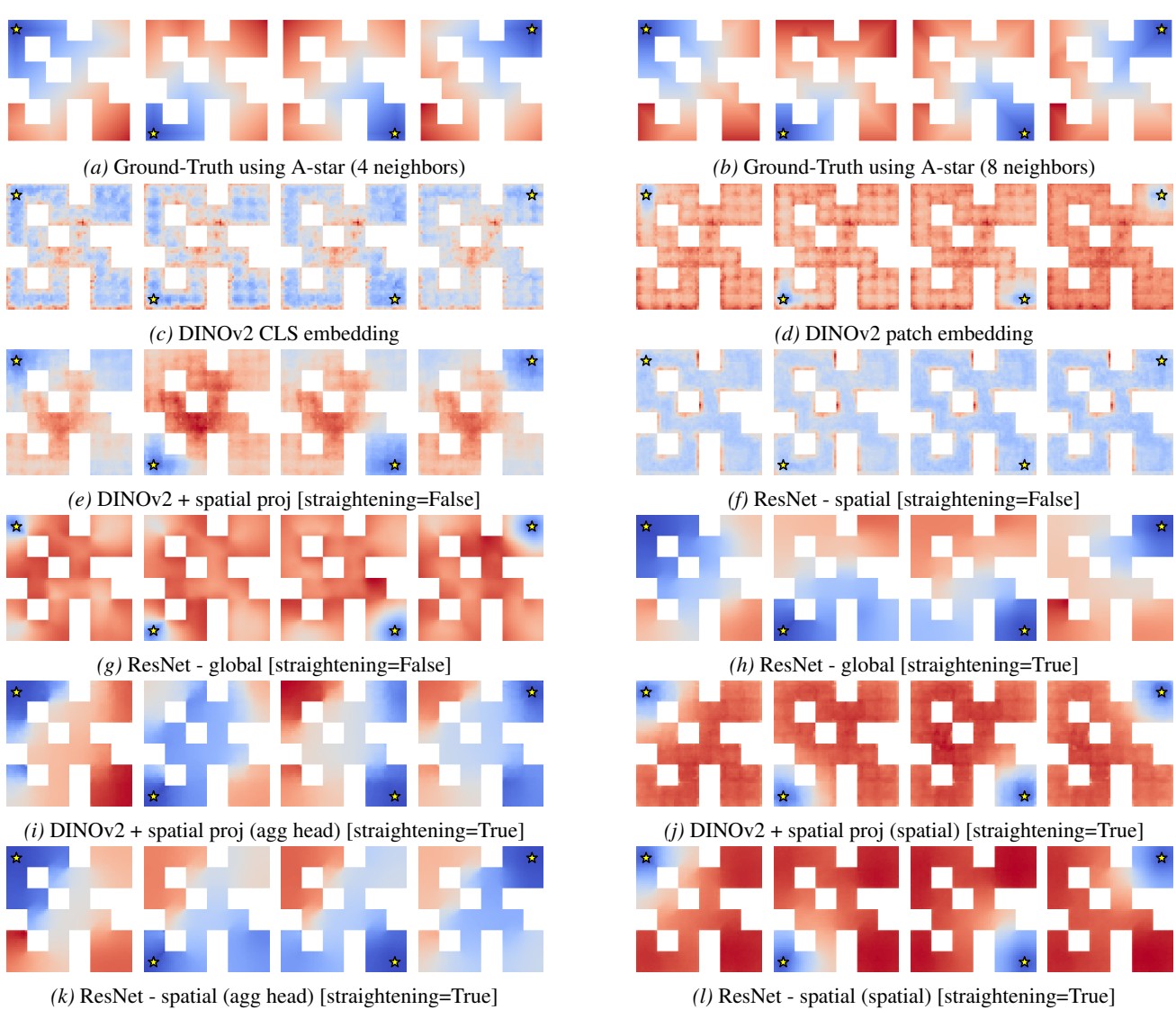

*(a)* Ground-Truth using A-star (4 neighbors)

*(b)* Ground-Truth using A-star (8 neighbors)

*(c)* DINOv2 CLS embedding

*(d)* DINOv2 patch embedding

*(e)* DINOv2 + spatial proj [straightening=False]

*(f)* ResNet - spatial [straightening=False]

*(g)* ResNet - global [straightening=False]

*(h)* ResNet - global [straightening=True]

*(i)* DINOv2 + spatial proj (agg head) [straightening=True]

*(j)* DINOv2 + spatial proj (spatial) [straightening=True]

*(k)* ResNet - spatial (agg head) [straightening=True]

*(l)* ResNet - spatial (spatial) [straightening=True]

*Figure 16.* Distance heatmaps of PointMaze-Medium.

### E.2. Visualization of Latent Trajectories

To visualize the learned representations of the trajectories, we randomly sample trajectories with a length of 30 and plot them in 2D using PCA. Here, we use DINO CLS token embeddings and the aggregated features of our model (trained with straightening). While latent trajectories are highly curved in DINO CLS embedding space, they become significantly smoother after straightening. Additionally, we compute the MSE between the embeddings of each intermediate state and the target. The Euclidean distance is closer to the geodesic distance for straighter trajectories, and thus MSE (which is squared Euclidean distance) becomes a more useful planning cost function that can reflect the true progress towards the target. Visualizations for different environments are in Figures 17 to 20.

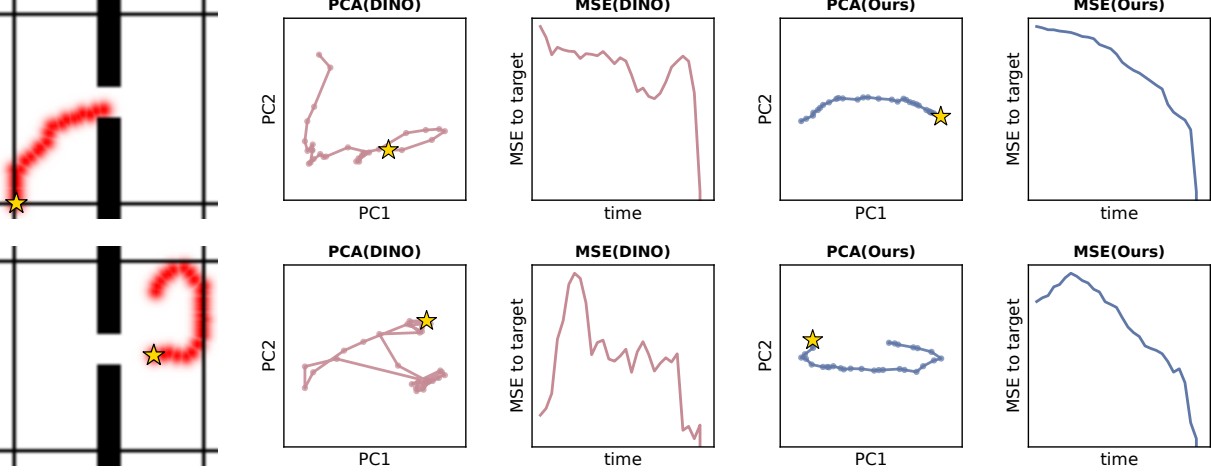

*Figure 17.* PCA of Trajectories of Wall.

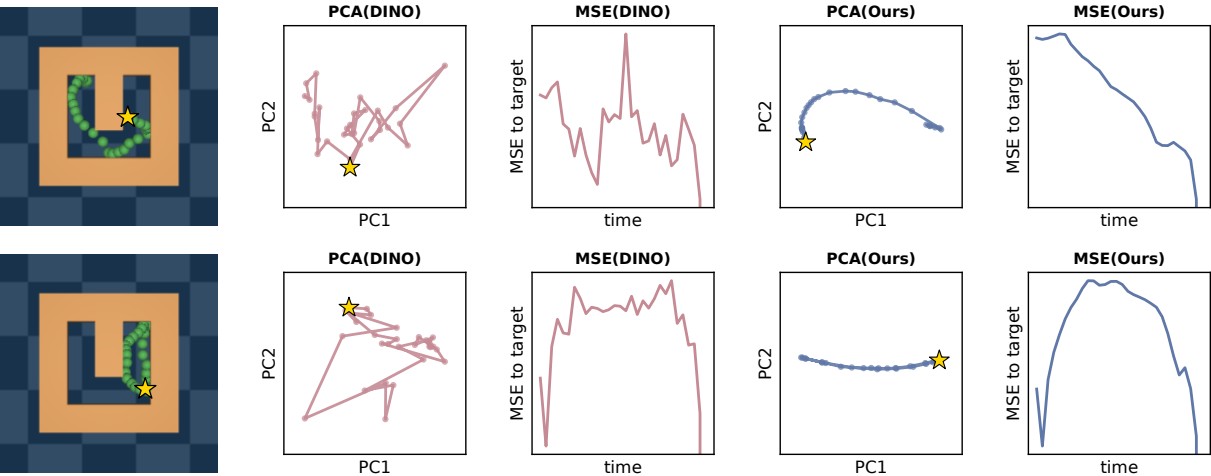

*Figure 18.* PCA of Trajectories of PointMaze-UMaze.

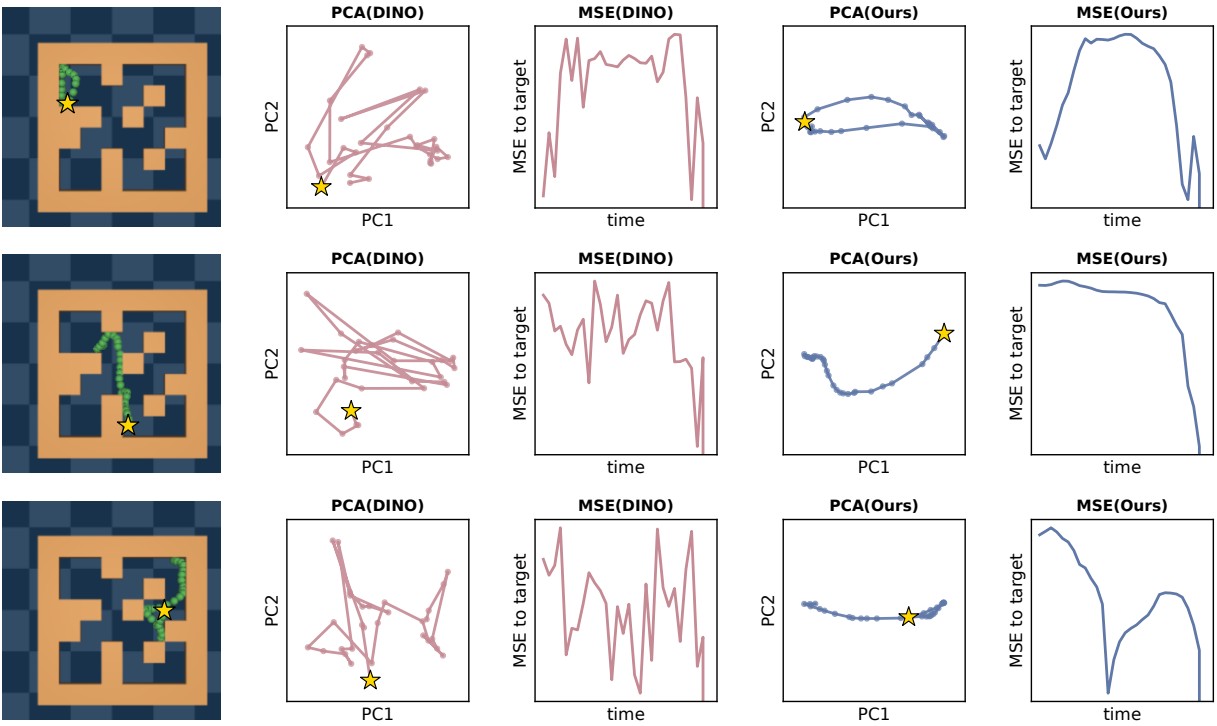

*Figure 19.* PCA of Trajectories of PointMaze-Medium.

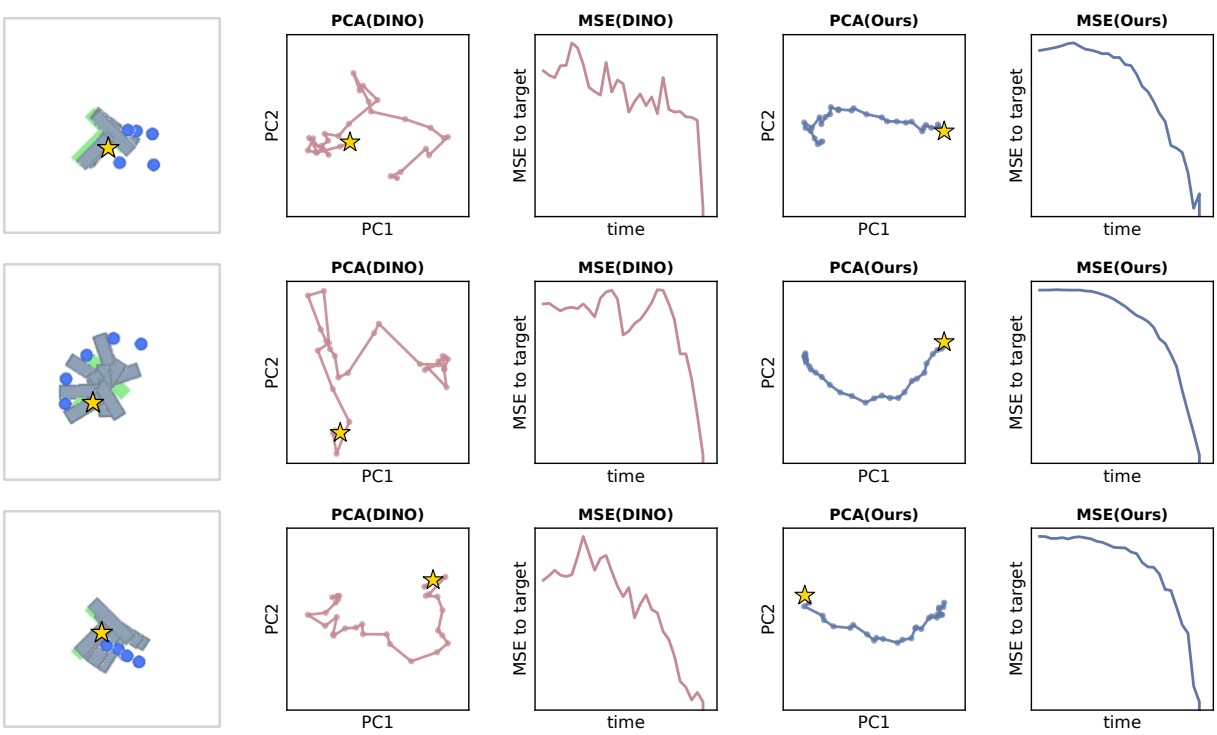

*Figure 20.* PCA of Trajectories of PushT. The overlaid figures only include five samples for readability.

## E.3. Planning Trajectories

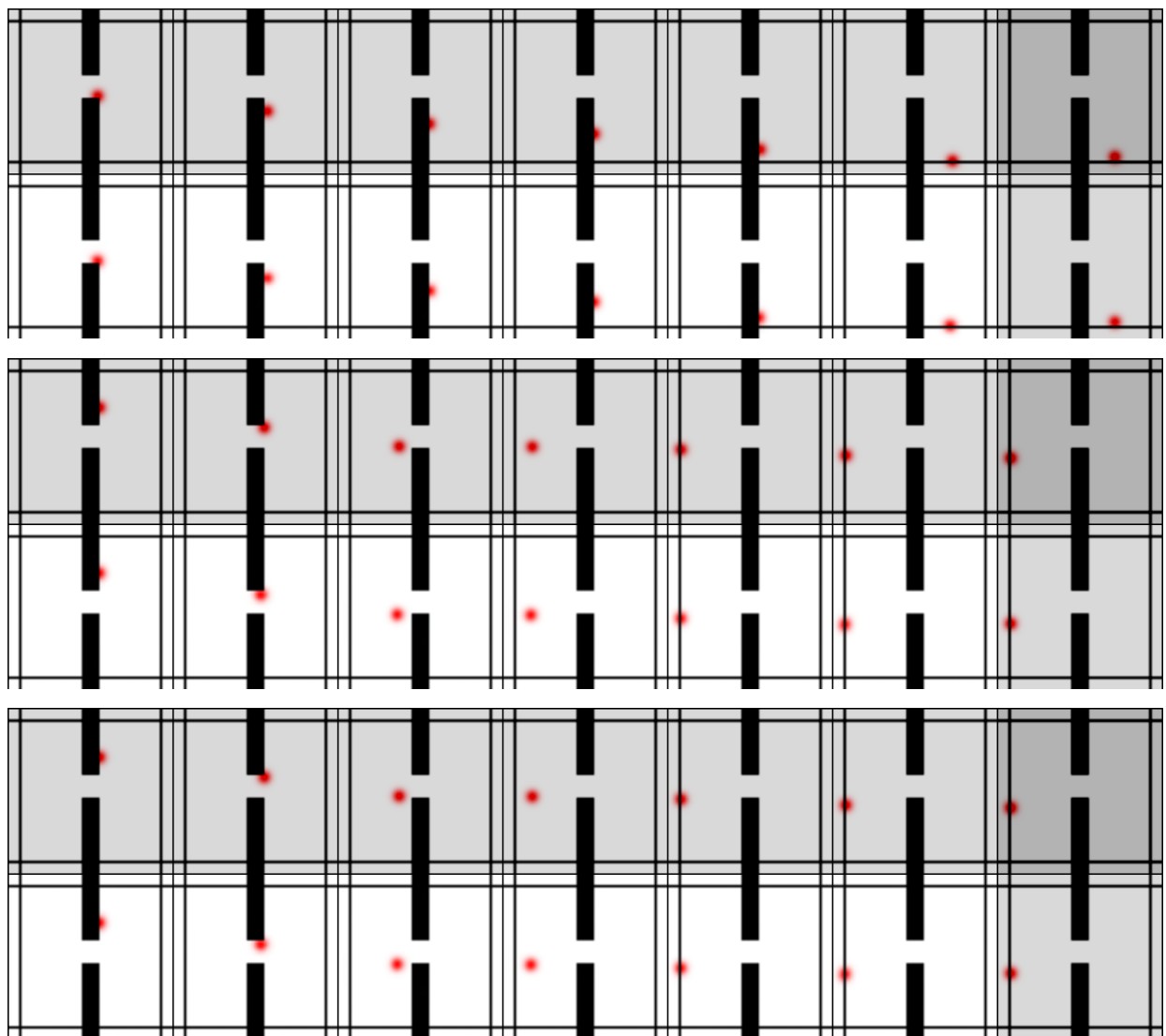

*Figure 21.* Open-Loop Planning Trajectories of Wall. The first row is from the simulator and the second from the decoder. The last column is the goal image.

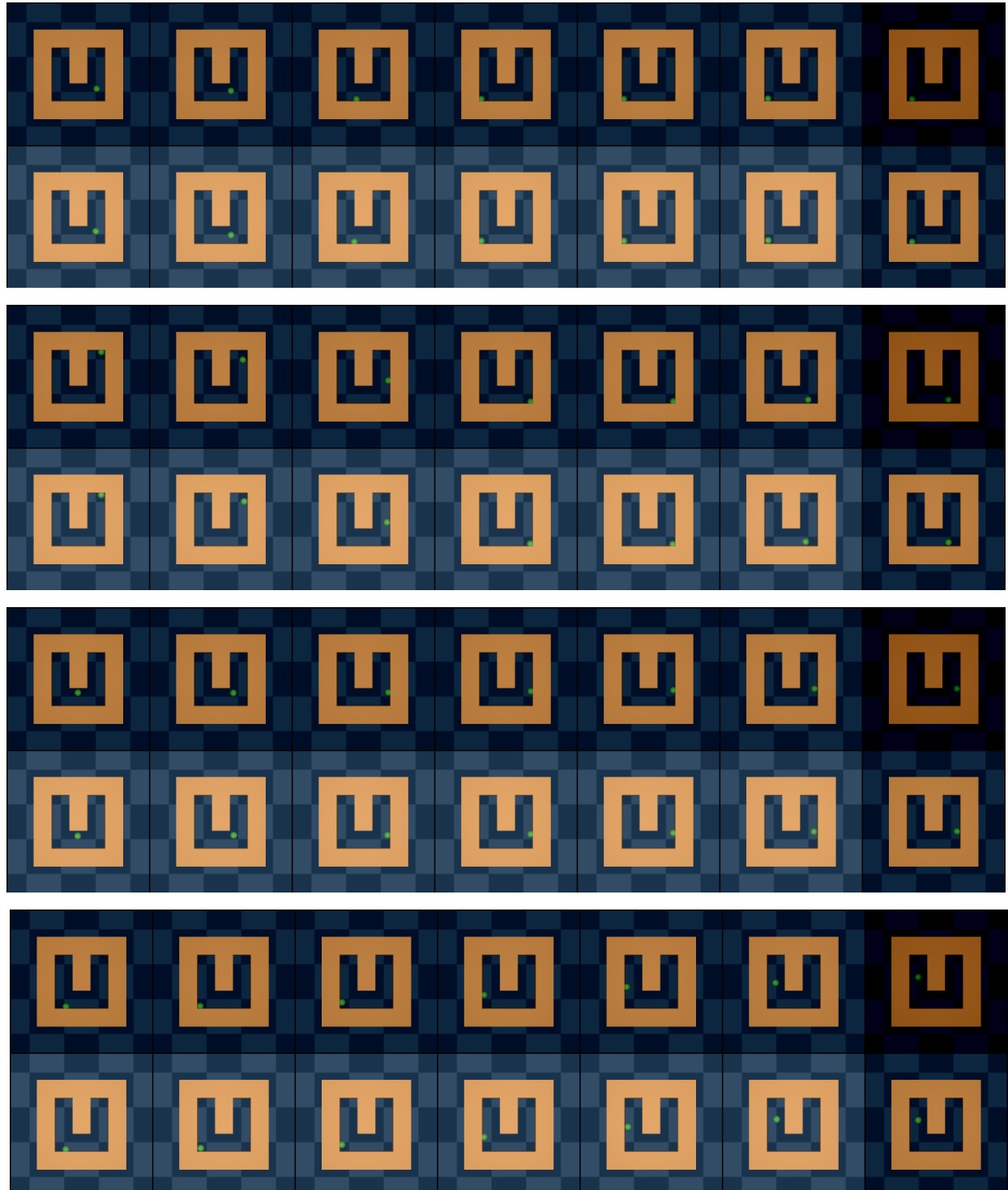

*Figure 22.* Open-Loop Planning Trajectories of PointMaze-UMaze. The first row is from the simulator and the second from the decoder. The last column is the goal image.

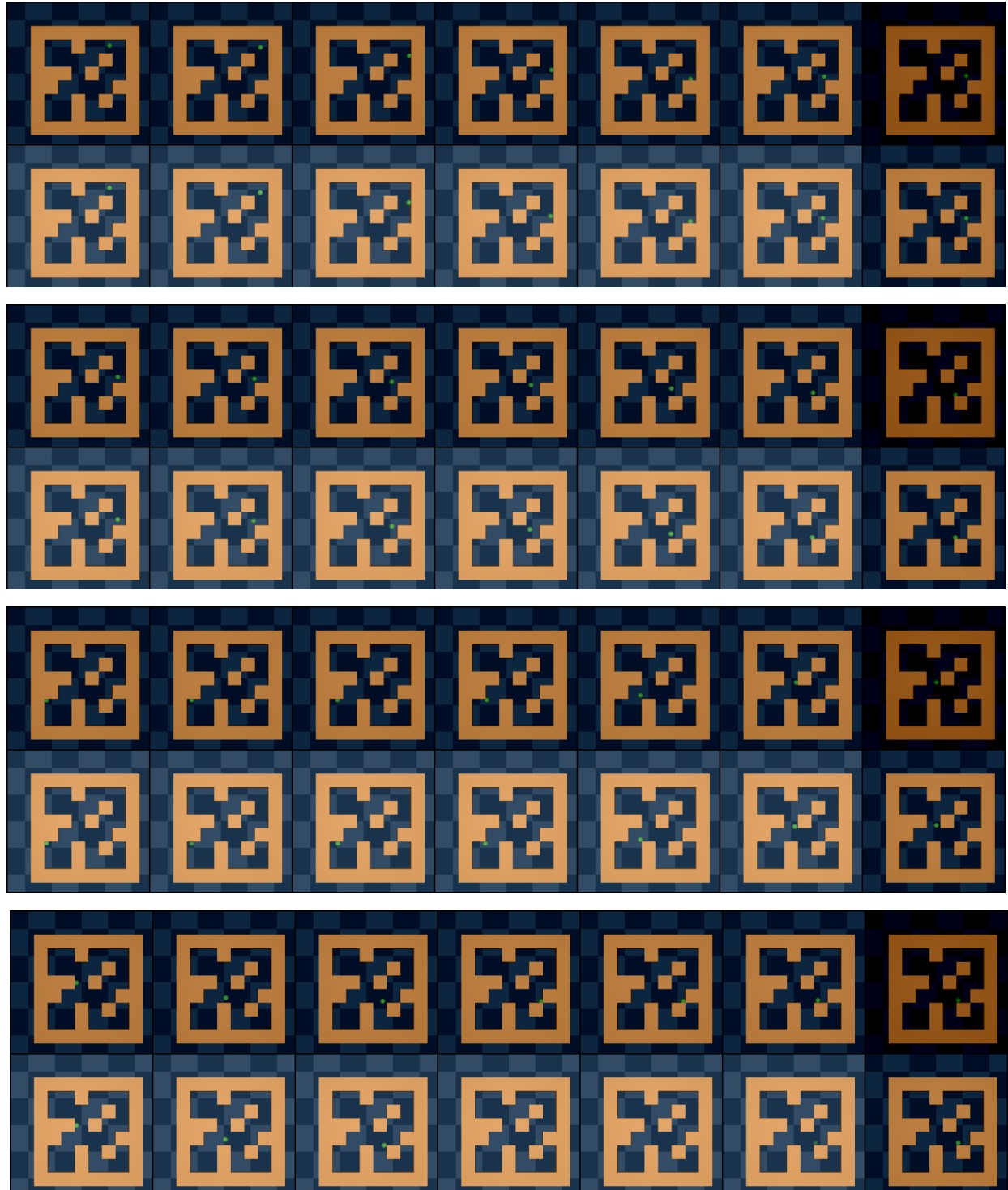

*Figure 23.* Open-Loop Planning Trajectories of PointMaze-Medium. The first row is from the simulator and the second from the decoder. The last column is the goal image.

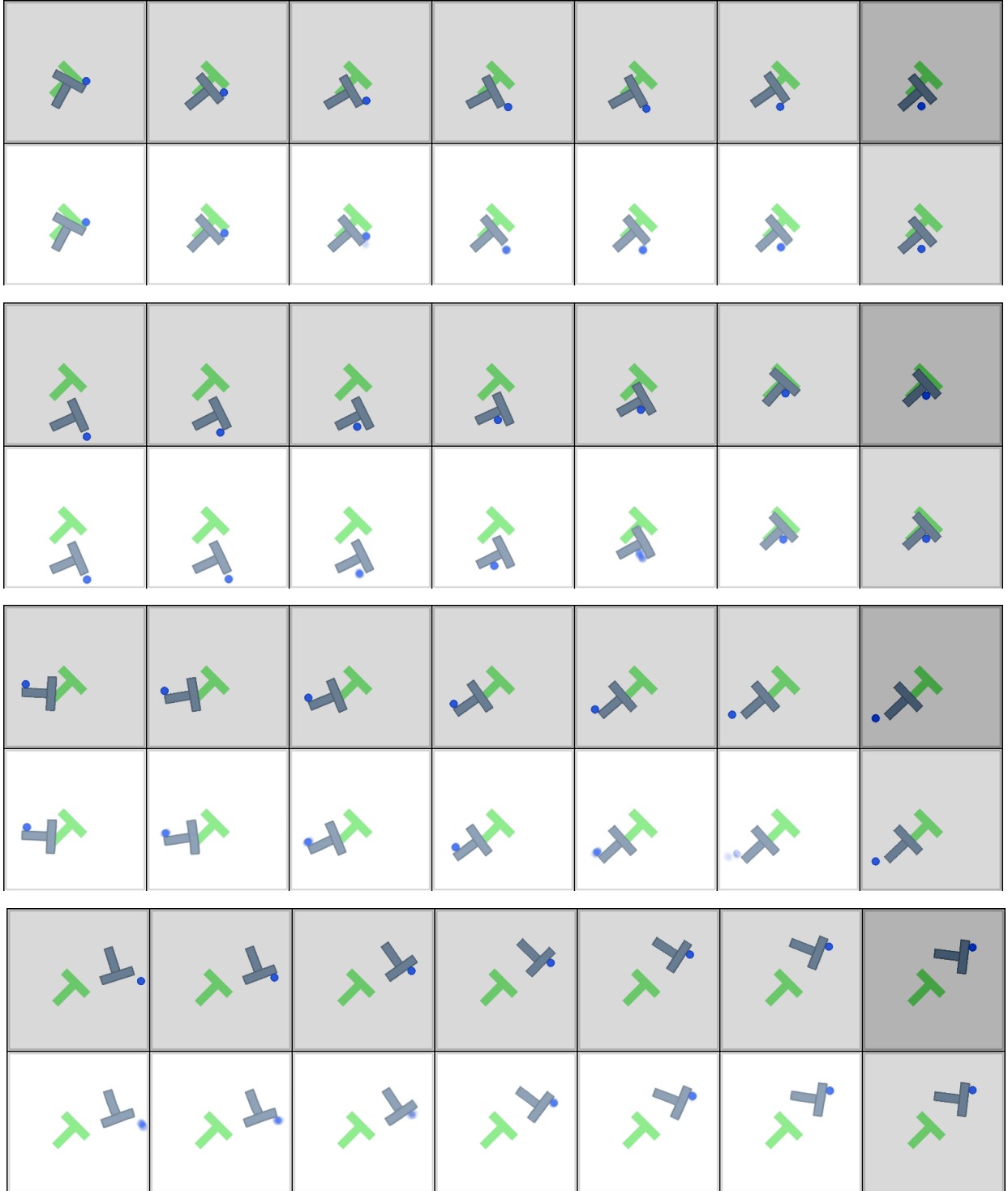

*Figure 24.* Open-Loop Planning Trajectories of PushT. The first row is from the simulator and the second from the decoder.

## F. Teleported PointMaze

This is a novel 2D navigation environment adapted from PointMaze. The core modification is a one-way teleportation dynamic. While the top, bottom, and left boundaries of the maze function as standard solid obstacles, a predefined region near the right wall acts as a teleportation trigger. If an agent's state transition at time $t$ results in a new x-position $x_{t+1}$ that crosses this threshold (i.e., $x_{t+1} > x_{\text{right-border}}$), an instantaneous state intervention occurs, modifying the agent's state as follows:

1. Position (x): The agent's x-position is reset to the left side of the maze: $x_{t+1} \leftarrow x_{\text{left-border}}$.

2. Position (y): The agent's y-position $y_{t+1}$ is preserved.

3. Velocity (x): The agent's x-axis velocity $v_x$ is reset to its absolute value: $v_{x,t+1} \leftarrow |v_{x,t}|$.

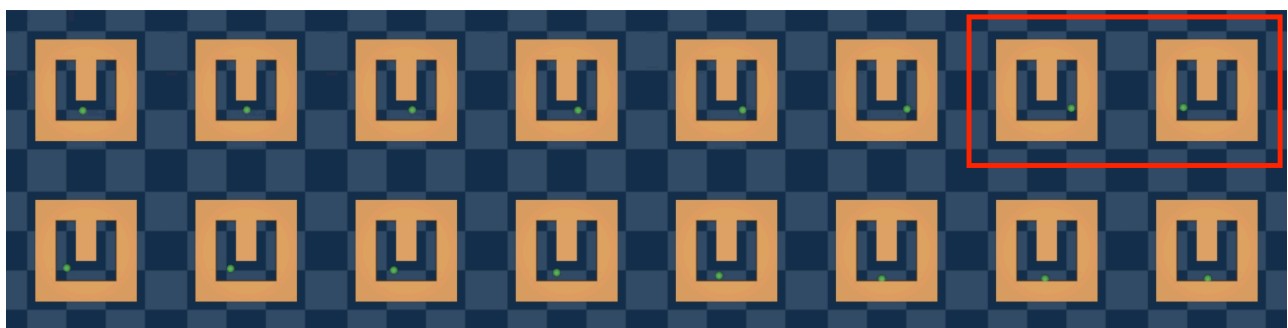

*Figure 25.* Teleported PointMaze. Note that the teleportation happens within the red box.

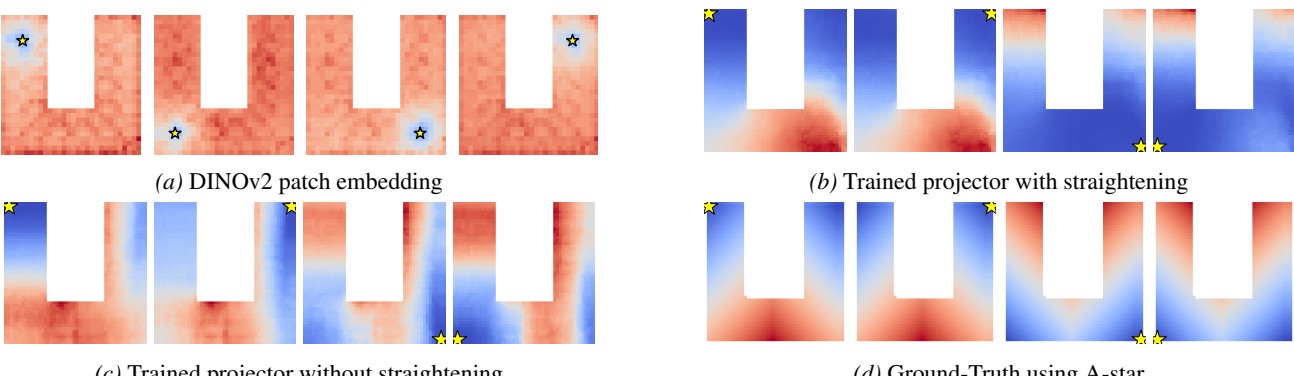

*(a)* DINOv2 patch embedding    *(b)* Trained projector with straightening

*(c)* Trained projector without straightening    *(d)* Ground-Truth using A-star

*Figure 26.* Distance heatmaps of Teleported PointMaze (blue indicates small values, red indicates large values). The state marked by the yellow star is used as the target, and we compute the MSE between its embedding and those of all other states in the maze. Since MSE is symmetric, this visualization does not fully capture directional reachability in the asymmetric teleportation dynamics. Nevertheless, with straightening, the resulting heatmaps are significantly closer to the transition-aware distances obtained using A-star.

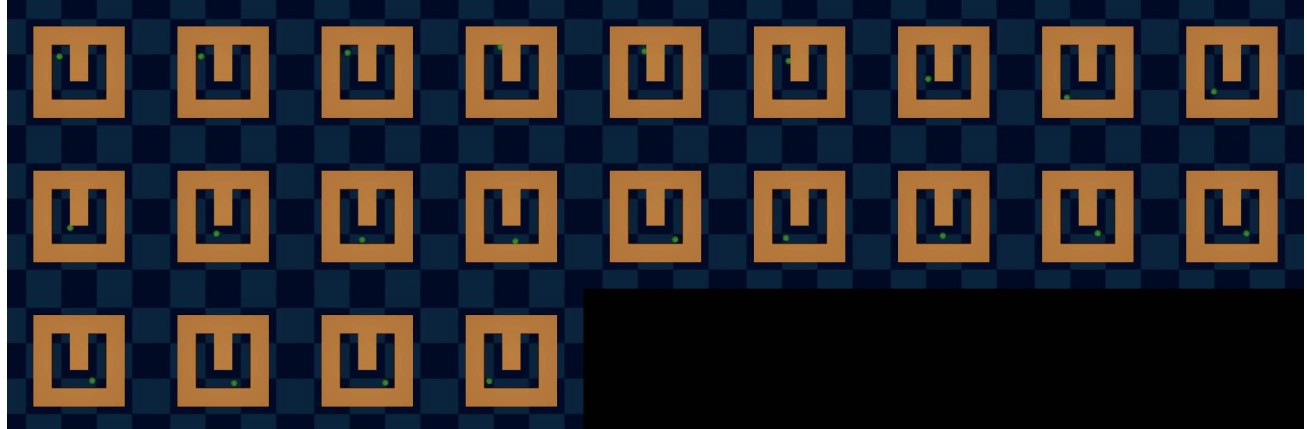

*(a)* With straightening, the agent reaches the target within given step limit.

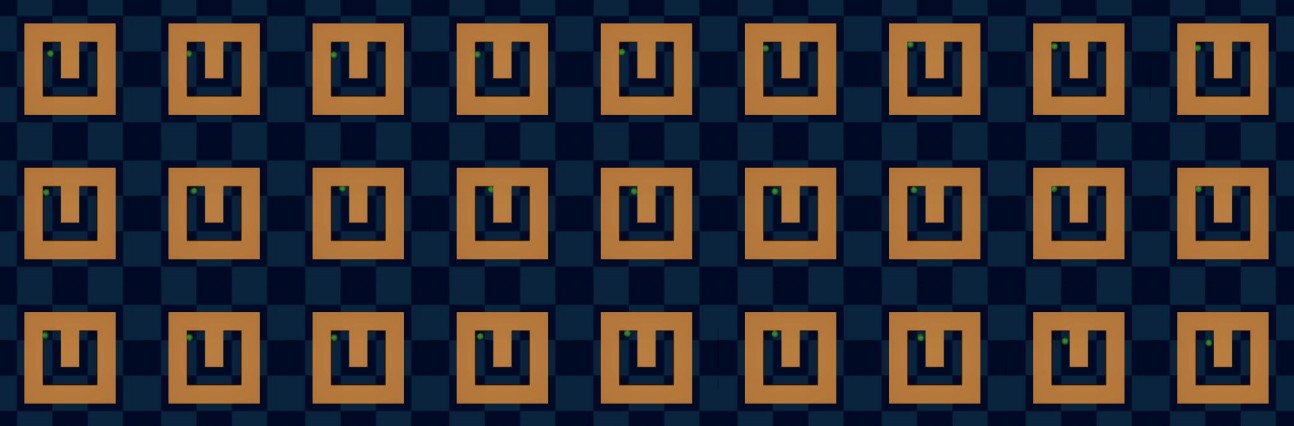

*(b)* Without straightening, the agent gets stuck at the corner.

*Figure 27.* Comparison of Planning Trajectories in Teleport-PointMaze. The frames were masked by black after reaching the target.

