# OpenReview forum: "Temporal Straightening for Latent Planning"
_ICML.cc/2026/Conference — ICML 2026 regular_

### Official Review · Reviewer_N2io · 2026-03-03

**Soundness:** 3
**Presentation:** 3
**Significance:** 3
**Originality:** 3
**Overall Recommendation:** 4
**Confidence:** 1

**Summary:**

This paper propose temporal straightening, a geometric regularizer for latent planning. It learns a small trainable projector on top of a frozen pretrained visual encoder (DINOv2) and add a temporal straightening regularizer that encourages consecutive latent “velocities” to be collinear. The world model is trained with a JEPA-style prediction loss plus this straightening regularizer. The experiments show the proposed method achieves promising results over the baselines on goal-reaching success rates for GD planning on Wall, PointMaze, and PushT.

**Compliance With Llm Reviewing Policy:**

Affirmed.

**Final Justification:**

I maintain my rating as stated below.

**Key Questions For Authors:**

Please refer to the weakness.

**Limitations:**

yes

**Strengths And Weaknesses:**

Strengths:
1. The idea of using a lightweight projector and a cosine-based straightening regularizer is easy to implement.
2. The authors provides a possible explanation for why straightening objective works via Hessian decomposition.
3. Experiments show promising results.

Weakness:
1. This paper lacks runtime comparisons with CEM or MPPI, since the motivation is to make gradient-based planning a practical alternative to CEM/MPPI, but the planning setup still uses horizon 25 and 100 optimization steps per replanning cycle.
2. The Comparisons are mostly conduct with direct planning using DINOv2 or a projector without straightening, so it's unclear how this compares to other planning-relevant representation-shaping methods like temporal contrastive objectives, smoothness or curvature regularizers, and anti-collapse schemes.
3. The evaluation is limited to four structured goal-reaching environments with offline data, raising questions about whether straightening would remain robust in more challenging settings—such as those involving partial observability, stochastic dynamics, contact-rich long-horizon tasks beyond PushT, or scenarios where the optimal latent dynamics are inherently curved.

---

> ### Author Rebuttal · Authors · 2026-03-31
>
> Thank you for the positive recommendation and helpful suggestions. We provide additional experiments and address each concern below.
>
> > This paper lacks runtime comparisons with CEM or MPPI
>
> We agree that the runtime comparison is critical and will include this in our revision. Note that GD/CEM are just different subplanners within the same open-loop/MPC setup, so the difference is only in the complexity of the subplanner. For one subplan:
> - GD: With N optimization steps, the cost is N forward and backward passes.
> - CEM: It iteratively samples M candidate action sequences, applies the learned predictor for rollout, refits a sampling distribution to top-performing candidates, and repeats this process for K iterations. The cost is MK forward passes, with M usually large for a competitive performance.
>
> We find that we need at least 200 samples x 10 iterations for CEM to achieve strong performance, resulting in **~10x slower** planning compared to GD. We report the wall-clock time of open-loop planning for 50 samples using one L40s GPU.
>
> |time (s) |GD (100 steps)  |CEM (200 samples x 10 iters)|
> | --- | --- | --- |
> |Global features (1x384) |8.1 $\\pm$ 1.4  |94.63 $\\pm$ 1.7  |
> |Patch features (196x384)|129.9 $\\pm$ 1.3 |1009.2 $\\pm$ 1.8 |
>
> A more comprehensive table on the tradeoff between performance and speed can be found in S1 of temporalstraightening.github.io. The model uses patch features trained with *straightening* regularization. **CEM requires large sample sizes for good performance, resulting in substantial latency, whereas GD can achieve competitive performance after straightening with significantly faster speed.**
>
> > [...] other planning-relevant representation-shaping methods like temporal contrastive objectives, smoothness or curvature regularizers, and anti-collapse schemes.
>
> Thanks for the suggestions. As requested, we experiment with temporal contrastive objectives and smoothness regularizations.
>
> - **Smoothness**: penalize large temporal jumps in visual embeddings $L_s= mean(||z_{t+1} - z_t||^2)$. Note that **a too large smooth penalty may result in a collapse** where embeddings of different states have the same value.
> - **Time contrastiveness**: Frames within window size k steps are positives, all others in the trajectory are negatives, encouraging temporally close embeddings to be similar and distant ones to differ $L_{tc} = InfoNCE(z_{pos}, z_{neg})$. However, as discussed in L98–101, when training trajectories are suboptimal, **this objective can incorrectly separate states that are geodesically close but temporally far apart**.
>
> We report the open-loop GD planning success rate on PushT, evaluated with 50 samples over 3 seeds. Overall, we do not observe improvements from adding smoothness or temporal contrastive objectives​. In fact, larger weights generally hurt performance, while smaller weights are less harmful but still do not match the gains from straightening. At the same time, we note that these objectives may be useful in other setups. Our method is also complementary to such objectives, and straightening can be combined with them.
>
> |  | SR (%) |
> | --- | --- |
> | L_pred | 66.00 $\\pm$ 5.66 |
> | L_pred + 0.1 \* L_straighten | **73.33** $\\pm$ 6.60 |
> | L_pred + 0.1 \* L_smooth | 28.67 $\\pm$ 3.40 |
> | L_pred + 0.01 \* L_smooth | 25.33 $\\pm$ 2.49 |
> | L_pred + 0.001 \* L_smooth | 65.33 $\\pm$ 4.11 |
>
> | | window size = 2  | window size = 5  |
> | --- | --- | --- |
> |L_pred + 0.1 \* L_tc  |48.00 $\\pm$ 7.12|43.33 $\\pm$ 4.11|
> |L_pred + 0.01 \* L_tc |33.33 $\\pm$ 2.49|42.00 $\\pm$ 5.89|
> |L_pred + 0.001 \* L_tc|60.00 $\\pm$ 4.90|61.33 $\\pm$ 6.60|
>
> We also compared four **variants of curvature implementations** for spatial features (also described in L238-244 in the paper): (i) patch: patch-wise cos and avg; (ii) flatten: flatten all patches and then cos; (iii) mean: average-pool patches then cos; (iv) agg: use a learnable pooling head to aggregate patches before cos. The results are in S2 of temporalstraightening.github.io. We find that (iv) performs best as straightening should act on the global trajectory representations, whereas patch tokens mainly capture local, patch-level variations that are only loosely aligned across time due to object motion and occlusion.
>
> As for anti-collapse, we simply use stop-grad because it doesn’t introduce additional hyperparameters or require negative samples.
>
> > The evaluation is limited to four structured goal-reaching environments [...]
>
> We add two challenging deformable object manipulation tasks to test our proposed methods further.
> - Rope: Move the rope to the goal configuration
> - Granular: Manipulate about a hundred particles to form desired shapes
>
> The results can be found under response to Reviewer YCas, which shows that **straightening consistently improves in these deformable object manipulation tasks**. We will include these results and continue to test in more challenging environments in future work.

---

> > ### Author Rebuttal · Reviewer_N2io · 2026-04-03
> >
> > The rebuttal  provides more experiments to answer my questions while I am not very familiar with this topic. Thus, I keep my positive rating.

---

> > > ### Author Response · Authors · 2026-04-04
> > >
> > > Thank you again for your positive recommendation! We added the experiments you suggested and hope they helped resolve your concerns. We would be happy to provide any further clarification or answer any follow-up questions.

---

### Official Review · Reviewer_YCas · 2026-03-09

**Soundness:** 3
**Presentation:** 2
**Significance:** 2
**Originality:** 3
**Overall Recommendation:** 4
**Confidence:** 3

**Summary:**

This paper aims to address the instability of gradient-based planning in latent space built on pre-trained visual encoders. It proposes a lightweight latent trajectory straightening method, which uses a trainable projector on top of a frozen pre-trained visual encoder paired with a curvature regularization loss.

**Compliance With Llm Reviewing Policy:**

Affirmed.

**Final Justification:**

All the concerns I raised have been fully addressed.

**Key Questions For Authors:**

1. Could you provide detailed explanations of the mechanism by which latent space straightening improves gradient-based planning, and justify the rationality of using cosine similarity as the curvature metric for the latent space?
2. Could you supplement performance comparisons with additional baselines such as Navigation World Models, DreamerV3, TD-MPC2, and CEM-based DINO-WM to verify the competitiveness of your method?
3. Could you add experimental results on more complex benchmarks such as Rope Manipulation and Granular Manipulation?
4. Could you provide theoretical analysis or empirical results of end-to-end latency to validate the claimed efficiency advantage of gradient-based planning over CEM, specifically verifying lower latency with comparable or better success rates?

**Limitations:**

The paper lacks further discussions about the applicability of the method in more complex environments, where straightening operations may harm the planning performance due to the more complex latent space. It also fails to elaborate on how the method can further improve success rates in longer-horizon settings.

**Strengths And Weaknesses:**

**strengths：**
1. The paper is overall well-written with a clear narrative.
2. The proposed method has meaningful novelty and practical engineering value. With temporal straightening, it effectively mitigates the instability of gradient descent-based planning and the tendency to fall into local optima.

**weakness：**
1. The paper lacks further mathematical explanations for why latent space straightening benefits planning, and does not validate the rationality of choosing cosine similarity as the curvature metric.
2. The paper primarily compares against the gradient descent version of DINO-WM, which lacks further comparisons with baselines such as Navigation World Models [1], DreamerV3 [2], TD-MPC2 [3], and the CEM-based DINO-WM [4].
3. Experiments are primarily conducted on the Wall, PointMaze, and PushT benchmarks, with no validation on more complex environments such as Rope Manipulation and Granular Manipulation.
4. The paper claims that gradient-based planning has lower computation cost and latency than sampling-based methods like CEM, but it doesn't provides detailed theoretical analysis or experimental results to support this claim.

[1] Bar, Amir, et al. Navigation world models.

[2] Hafner, D., Pasukonis, J., Ba, J., and Lillicrap, T. Mastering diverse domains through world models.

[3] Td-mpc2: Scalable, robust world models for continuous control.

[4] Zhou, Gaoyue, et al. Dino-wm: World models on pre-trained visual features enable zero-shot planning.

---

> ### Author Rebuttal · Authors · 2026-03-31
>
> Thank you for the constructive review. We are glad you found our method **novel with practical value**. We address each point below.
>
> > justification for straightening improves GD & cosine similarity
>
> We have included an analysis of how straightening improves GD convergence via Hessian decomposition in sec4, with an empirical example of loss landscape becoming closer to convex after straightening in fig3. We now provide a formalized version under a simplified linear latent dynamics and will add a more detailed proof in our revision.
>
> ---
>
> Let the latent dynamics be $z_{t+1}=Az_t+Ba_t$ and the planning objective over horizon K be $\mathcal{L}(\mathbf{a})=\\|z_K(\mathbf{a})-z_g\\|_2^2.$ Unrolling gives
>
> $$
> z_K(\mathbf{a})=A^K z_0 + G\mathbf{a},
> \qquad
> G=[A^{K-1}B,\\;A^{K-2}B,\dots,B],
> $$
>
> so the Hessian is $ H=\nabla_{\mathbf{a}}^2 \mathcal{L}(\mathbf{a})=2G^\top G$. Let $W=GG^\top=\sum_{t=0}^{K-1}A^tBB^\top(A^t)^\top$ be the finite-horizon controllability Gramian. Hence, GD convergence is governed by the condition number of $W$ (equivalently $H$).
>
> We define **$\varepsilon$-straight**: $\\|A-I\\|\_2\le\varepsilon$ and assume square action with B invertible. Then $\\|A\\|\_2\\le 1+\\varepsilon$ and $\\sigma_{\min}(A) \\ge 1-\\varepsilon$, which implies
>
> $$
> \\kappa\_{\mathrm{eff}}(H)\\le
> \\kappa(B)^2
> \\frac{\sum\_{t=0}^{K-1}(1+\varepsilon)^{2t}}
> {\\sum\_{t=0}^{K-1}(1-\varepsilon)^{2t}}.
> $$
>
> Thus, when $\varepsilon=0$, the condition number is $O(1)$ in $K$. This formalizes that **straightening yields a better-conditioned planning objective, which translates to faster convergence**.
>
> ---
>
> For the curvature metric, define $C_t:=\cos(v_t,v_{t+1})$ where $v_t=z_{t+1}-z_t$. Under linear dynamics,
>
> $$
> v_{t+1}=Av_t+B(a_{t+1}-a_t),
> \qquad
> (A-I)v_t=(v_{t+1}-v_t)-B(a_{t+1}-a_t).
> $$
>
> Let $\hat v_t=\frac{v_t}{\\|v_t\\|_2}$. Assume constant latent speed $\\|v_t\\|_2=c$ and smooth actions $\Delta_a:=\max_t\\|a\_{t+1}-a\_t\\|_2 < \infty$, then
>
> $$
> \\|(A-I)\hat v\_t\\|_2
> \le
> \sqrt{2(1-C\_t)}+\frac{\sigma\_{\max}(B)\Delta\_a}{c}.
> $$
>
> Thus, **high cosine similarity implies the transition is close to identity along visited directions, which explains why maximizing cos serves as a useful proxy for straightness**.
>
> > Navigation World Models, DreamerV3, TD-MPC2, and CEM-based DINO-WM
>
> Thanks for the suggestion! We did not include DreamerV3 or TD-MPC2 because they require reward supervision during training while our setting is **reward-free**. DINO-WM (Zhou et al., 2025) already benchmarks these methods in the reward-free setting: On PushT, DreamerV3 achieves 30% success rate while TD-MPC2 achieves 0. We have also attempted to adapt the offline variant of TD-MPC2 to our tasks ourselves using the exponential of the negative state-space distance as a reward proxy, but still observed very low success rates (0 for PushT and 6% for PointMaze) after training for 1M steps.
>
> NWM is also not comparable as it is a 1B-parameter video model for navigation trained on large-scale egocentric video, whereas our predictor is only **~1.4M parameters, operates in latent space, and does not generate videos**. Straightening could potentially benefit NWM as well, but we are unable to test it during rebuttal due to the compute limit (NWM requires 64 H100s).
>
> For the most direct comparison, we report open-loop CEM results in S1 in temporalstraightening.github.io. **GD after straightening can already surpass CEM before straightening. Moreover, straightening also improves CEM in most environments**
>
> > rope & granular
>
> Following your suggestions, we test our proposed methods on rope and granular. We report the average final Chamfer Distance (CD) below, using MPC with GD planner. MPC planning examples can be found under S4 of temporalstraightening.github.io. **Straightening consistently improves in these challenging deformable object manipulation tasks**.
>
> ||straighten|Rope|Granular|
> |---|---|---|---|
> |DINO-WM|- |1.08 |0.46|
> |+proj |F|0.95|0.24|
> |+proj |T |**0.74**|**0.19**|
>
> > GD vs CEM
>
> Please find a detailed comparison under our response to reviewer N2io. Overall, **GD is able to achieve better performance-speed tradeoff with straightening**.
>
> > straightening in more complex environments & longer-horizon setting
>
> We agree that over-straightening can hurt prediction and planning, but the regularization strength is tunable. To avoid forcing all patch features to be straight, we also use a learnable pooling head and apply straightening only to the pooled features, which reduces global curvature while leaving local patch features less constrained (as discussed in L324–328).
>
> For longer horizon, we have included a 50-step setting in Tab2 in the paper. While all methods degrade with horizon, straightening consistently improves over the baseline. We also observe that the pooled features learned with straightening provide a more coherent long-range distance-to-goal signal, enabling even longer-horizon planning (S3 in temporalstraightening.github.io).

---

> > ### Author Rebuttal · Reviewer_YCas · 2026-04-03
> >
> > All the concerns I raised have been fully addressed, and I will raise my score accordingly.

---

> > > ### Author Response · Authors · 2026-04-04
> > >
> > > Thank you for your positive feedback! We are glad that our rebuttal addressed your concerns, and we will incorporate the additional discussion and experiments in the revision.

---

### Official Review · Reviewer_tnjX · 2026-03-13

**Soundness:** 3
**Presentation:** 3
**Significance:** 3
**Originality:** 3
**Overall Recommendation:** 5
**Confidence:** 4

**Summary:**

This paper introduces a novel loss for latent dynamics learning.
Given a trajectory, it tries to minimize the steering in the latent space.
This smoothes out the latent space geometry and makes the loss landscape of latent goal distance convex.
This is a nice combination of theoretical intuition and the empirical result.

The proposed idea is, to my knowledge, highly novel.
However, the scope and the impact of the work is limited to simple low-level control domains.

**Compliance With Llm Reviewing Policy:**

Affirmed.

**Final Justification:**

The author promised to clarify the scope of the paper in the camera-ready.

**Key Questions For Authors:**

Do you agree that this method is inherently limited to the symmetric dynamics?
If not, how does it address it?

**Limitations:**

I believe this method is inherently limited to the symmetric dynamics, and the paper does not evaluate such a potential failure scenario.

**Strengths And Weaknesses:**

The method is evaluated on a fairly simple control domain such as maze, wall, pushT, etc.,
that does not require planning, i.e., deliberation into the future by systematically exploring the alternatives.
As such, the paper title should be changed to "... latent control".

The success of the proposed approach is predicated by
the relatively unconstrained environment with a uniform cost function.
While this applies to a large subset of robotic low-level planning,
it does not generalize beyond that.

For example, how it would behave if the environment dynamics is asymmetric, e.g., an
environment with a strong wind that acts as a one-way corridor?
A gradient-based planner that learned from the trajectories that goes along the wind
would keep trying to approach the goal on
the shortest geodesic path even if it is against the wind, and fail in practice.
A task that can be solved by a reflex agent which simply performs a hill-climbing on a loss landscape
does not require the ability to plan, which is to foresee the future and search for a successful alternative.

Another example is the task-based planning.
Latplan [Asai et. al, JAIR2020] learns a propositional, i.e., discrete latent space from images
with discrete latent actions. What would be the smothness in such a discrete space?
A similar assymmetric analogy also works here.
If you empty a can of soda, there is no going back --- if you need a full can again,
the only way to achieve it is to go to the fridge and pop a new one.
There is no way to return the environment to the past state unlike in the simple domains evaluated in this work.

The training method also lacks the statistical interpretation.
Every regularization loss should be justified as a term that appears in some lower bounds of the likelihood,
derived from a hierarchical generative model, a variational model, or a density-ratio estimator.
In this work in particular, it would eventually be based on an extension of hidden markov model but I am not sure.

---

> ### Author Rebuttal · Authors · 2026-03-31
>
> Thank you for the positive recommendation and especially for recognizing the **novelty** of our idea. We appreciate your thoughtful and constructive review, and we address each point below.
>
> > The method is evaluated on a fairly simple control domain such as maze, wall, pushT, etc., that does not require planning, i.e., deliberation into the future by systematically exploring the alternatives.
>
> We agree that our experiments focus on continuous, goal-conditioned planning/control, and we will clarify this scope more explicitly. In our setting, the model is trained only on random/suboptimal reward-free trajectories. At test time, the agent is given an arbitrary goal and must **optimize an action sequence over a horizon using a learned dynamics model to minimize a goal-dependent cost specified only at test**. In this sense, planning is still necessary even in these environments as success requires selecting among alternative action sequences based on predicted future outcomes. We further show the success of our method in **longer-horizon maze navigation and challenging deformable object manipulation** tasks in S3–4 in temporalstraightening.github.io. We therefore believe “latent planning” is an appropriate term for the problem studied here. More broadly, “planning” is also used in similar settings in prior work such as
> - Learning Latent Dynamics for Planning from Pixels (ICML2019)
> - Learning and Planning in Complex Action Spaces (ICML2021)
> - DINO-WM: World Models on Pre-trained Visual Features enable Zero-shot Planning(ICML2025)
>
> We hope this clarifies our use of the term “planning,” and we would be happy to further clarify if the reviewer still has concerns.
>
> > For example, how it would behave if the environment dynamics is asymmetric, e.g., an environment with a strong wind that acts as a one-way corridor? [...] Do you agree that this method is inherently limited to the symmetric dynamics?
>
> Asymmetric dynamics is a challenging case, but the issue lies primarily in the symmetric distance objective, not in straightening itself. When reachability is directional, a symmetric L2 distance is not the right target and quasimetrics can be more appropriate as they can represent directional costs and irreversible dynamics. **Our straightening regularization penalizes curvature along observed transitions, rather than assuming reversibility or symmetric shortest paths.** Under asymmetric dynamics, it can still help and combined with quasimetric goal costs, and we believe this will be a very interesting future extension. We will include this discussion in our revision.
>
>
> > Another example is the task-based planning. Latplan [Asai et. al, JAIR2020] learns a propositional, i.e., discrete latent space from images with discrete latent actions. What would be the smothness in such a discrete space?
>
> Using a discrete latent space with discrete latent actions is a design choice rather than a requirement of the planning problem itself. Our method is orthogonal to that line of work: we focus on continuous latent representations with differentiable rollouts, and evaluate on continuous-action tasks. In this setting, straightness is a well-defined geometric property of latent trajectories. We will clarify the scope in the revision.
>
> > The training method also lacks the statistical interpretation. Every regularization loss should be justified as a term that appears in some lower bounds of the likelihood, derived from a hierarchical generative model, a variational model, or a density-ratio estimator. In this work in particular, it would eventually be based on an extension of hidden markov model but I am not sure.
>
> Our objective is intended as a geometric regularizer for planning, rather than a likelihood-derived term. For that reason, we think the most direct justification is geometric-based rather than probabilistic. Under our response to YCas, we provide the justification of straightening under a simplified linear dynamics setting: we connect straightness to the conditioning of the planning Hessian, and explain why cosine similarity between consecutive latent velocities is a practical proxy for straightness. We will incorporate these theoretical analysis in the paper.

---

> > ### Author Rebuttal · Reviewer_tnjX · 2026-04-06
> >
> > I've read the rebuttal.
> >
> > I believe the novelty in this paper deserves a credit even with its limited scope, and it should be accepted over other boring papers in my batch. However, the scores of other reviewers might not be strong enough to ensure acceptance, so I raise my score to secure it.
> > However, this is based on a huge trust on the author's promise to update the paper to acknowledge the limintation in the final camera-ready.

---

> > > ### Author Response · Authors · 2026-04-07
> > >
> > > Thank you for your support and positive feedback! We truly appreciate your recognition of the novelty of our work. We will make sure to discuss the limitations clearly in the revision.

---

### Decision · Program_Chairs · 2026-04-30

**Decision:**

Accept (regular)

**Comment:**

This paper proposes temporal straightening, a geometric regularizer for gradient-based planning in latent space that reduces trajectory curvature via a projector and cosine-similarity loss. All three reviewers (scores: 5, 4, 4) recognize the novelty and practical value, with rebuttals addressing concerns about limited scope, baseline comparisons, and theoretical grounding. The paper provides solid empirical results on goal-reaching tasks.   OVerall, this is a technically sound contribution with clear novelty that will interest the world models and planning community, so "ACCEPT".